# TIB: Detecting Unknown Objects via Two-Stream Information Bottleneck

## Abstract

Detecting diverse objects, including ones never-seen-before during model training, is critical for the safe application of object detectors. To this end, a task of unsupervised out-of-distribution object detection (OOD-OD) is proposed to detect unknown objects without the reliance on an auxiliary dataset. For this task, it is important to reduce the impact of lacking unknown data for supervision and leverage in-distribution (ID) data to improve the model's discrimination ability. In this paper, we propose a method of Two-Stream Information Bottleneck (TIB), which consists of a standard Information Bottleneck and a dedicated Reverse Information Bottleneck (RIB). Specifically, after extracting the features of an ID image, we first define a standard IB network to disentangle instance representations that are beneficial for localizing and recognizing objects. Meanwhile, we present RIB to obtain simulative OOD features to alleviate the impact of lacking unknown data. Different from standard IB aiming to extract task-relevant compact representations, RIB is to obtain task-irrelevant representations by reversing the optimization objective of the standard IB. Next, to further enhance the discrimination ability, a mixture of information bottlenecks is designed to sufficiently capture object-related information. In the experiments, our method is evaluated on OOD-OD and incremental object detection. The significant performance gains over baselines show the superiorities of our method.

## 1 Introduction

With the rejuvenation of deep neural networks, for object detection, many advances Ren et al. (2015); Redmon et al. (2016); Carion et al. (2020); Chen et al. (2022) have been achieved. Most existing methods often follow a close-set assumption that the training and testing processes share the same category space. However, the practical scenario is open and filled with unknown objects, presenting significant challenges for object detectors trained based on the close-set assumption. To this end, a task of unsupervised out-of-distribution object detection (OOD-OD) Du et al. (2022b) is recently proposed, whose goal is to accurately detect the objects never-seen-before during training without accessing any auxiliary data. Obviously, addressing this task is helpful for promoting the safe deployment of object detectors in real scenes, e.g., autonomous driving.

The main challenge of unsupervised OOD-OD is lacking supervision signals from OOD data during training Du et al. (2022b). In particular, as shown in the left part of Fig. 1, an object detector is typically optimized only based on the in-distribution (ID) data. During inference, the detector could accurately localize and recognize ID objects but easily produces overconfident incorrect predictions for OOD objects. The reason is that the object detector could not learn a clear discrimination boundary between ID objects and OOD objects in the case of lacking OOD data for supervision. Thus, for this task, one feasible solution is to extract simulative OOD data based on the ID data. And the simulative OOD data could be used to improve the discrimination ability of the object detector.

In order to obtain simulative OOD data, it is general to leverage generative methods, e.g., generative adversarial networks Lee et al. (2018a) and mixup Zhang et al. (2018), to synthesize OOD images. Though these methods have been demonstrated to be effective, using a large number of synthesized images may increase computational costs. Meanwhile, it is difficult to use synthesized images to cover the overall object space, which may weaken the discrimination performance for certain unknown objects.

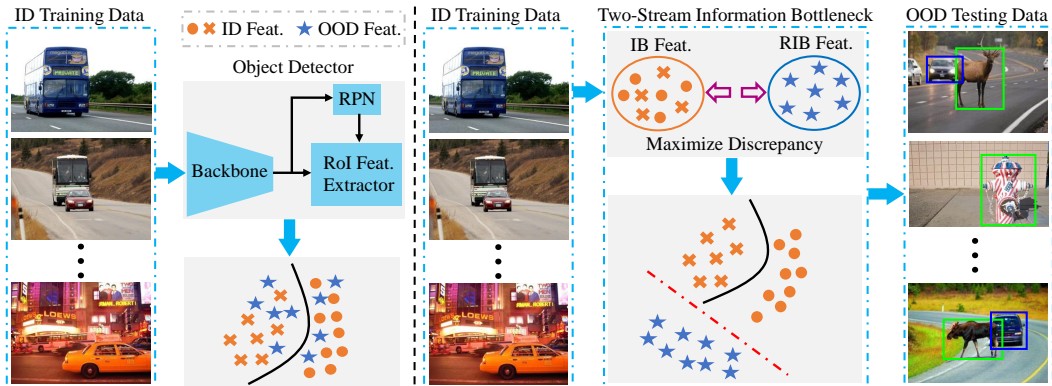

Figure 1: Two-Stream Information Bottleneck for OOD-OD. 'RPN' is Region-Proposal Network with RoI Alignment. The green boxes are OOD objects. The red and black lines separately indicate the decision boundary between ID and OOD objects and that between ID objects belonging to different categories. Due to lacking unknown data for supervision, the traditional object detector could not distinguish ID objects from OOD objects effectively. Our method aims to generate simulative OOD features by maximizing the prediction discrepancy between the features extracted by the IB module and that extracted by the RIB module, which enhances the discrimination ability.

In this paper, we explore employing Information Bottleneck (IB) Tishby et al. (2000); Alemi et al. (2017) to obtain a series of simulative OOD features for training. Particularly, we propose a method of Two-Stream Information Bottleneck (TIB) to improve the discrimination ability of the object detector, which mainly consists of a standard Information Bottleneck and a dedicated Reverse Information Bottleneck (RIB). Specifically, as shown in the right part of Fig. 1, given an ID image as the input, a backbone network, e.g., ResNet He et al. (2016), is used to extract the corresponding representations. Then, a standard variational IB Alemi et al. (2017) is defined to decompose an Instance map from the backbone representations, which is instrumental in localizing and recognizing objects accurately. Besides, standard IB struggles to extract maximally compressed features of the input while preserving as much task-relevant information as possible Lee et al. (2021). Whereas, OOD features could be considered irrelevant to the current task. Thus, we present RIB to obtain an OOD map used to extract task-irrelevant representations via reversing the optimization objective of the standard IB. Concretely, by maximizing the discrepancy between the predictions from the Instance map and that from the OOD map, and simultaneously minimizing the classification loss, the OOD map could be promoted to contain plentiful object-irrelevant information, which is beneficial for extracting simulative OOD features and improves the discrimination ability.

Furthermore, recent research Schulz et al. (2020) has shown that IB is an effective mechanism to capture object information. Inspired by this idea, we explore designing a mixture of information bottlenecks to purify object-related information from multiple different facets. Finally, by combining the information, the discrimination ability could be further enhanced. In the experiments, our method is separately evaluated on OOD-OD and incremental object detection Kj et al. (2021). Extensive experimental results demonstrate the superiorities of our method.

The contributions of our work are summarized as follows:

- We propose a method of Two-Stream Information Bottleneck consisting of a standard IB and a dedicated RIB. Particularly, RIB aims to obtain simulative OOD features by maximizing the prediction discrepancy between ID features and OOD features, which reduces the impact of lacking unknown data for supervision.

- We design a mixture of information bottlenecks to purify object-related information from multiple different facets, which is beneficial for enhancing object-related information in the features for classification and improves the detection performance.

- Experimental results show that our method could effectively improve the performance of OOD-OD and incremental object detection. Particularly, for PASCAL VOC Everingham et al. (2010), compared with the baseline method Du et al. (2022b), our method significantly reduces FPR95 by around 10.42%.

## 2 RELATED WORK

**OOD detection.** To promote the safe application of models in practical scenarios, OOD detection Pimentel et al. (2014); Yang et al. (2021b) has attracted much attention, whose goal is to distinguish ID data from OOD data. Most existing methods Lee et al. (2018b); Hendrycks et al. (2019); Lee et al. (2018a); Liang et al. (2017); Lee et al. (2018b) focus on OOD detection for image classification and exploit a regularization operation or an auxiliary dataset to address this problem. Particularly, Bendale et al. Bendale & Boult (2016) developed the OpenMax score for OOD detection based on the extreme value theory. Meanwhile, there exist some methods that explore utilizing temperature scaling Liang et al. (2017), generative models Lee et al. (2018b); Serrà et al. (2019), or ensemble methods Vyas et al. (2018); Choi et al. (2018) to calibrate the distribution of the softmax score. Besides, Liu et al. Liu et al. (2020) proposed to leverage the energy-based idea to address OOD detection, which opens a new solution. Yang et al. Yang et al. (2021a) and Zhou Zhou (2022) separately leveraged the idea of semantically coherent and that of reconstruction to distinguish OOD data. Though these methods have been shown to be effective, since object detection involves object localization and recognition, these methods could not be directly used for OOD-OD.

Recently, unsupervised OOD-OD Du et al. (2022b) is proposed to determine whether detected objects belong to out-of-distribution or not without accessing an auxiliary dataset. For this task, Du et al. Du et al. (2022b) proposed to synthesize virtual outliers to reduce the impact of lacking unknown data for supervision. The work Du et al. (2022a) explores learning more unknown-related knowledge from an auxiliary video dataset, which could not be used for unsupervised OOD-OD. Harakeh et al. Harakeh & Waslander (2021) mainly focused on uncertainty estimation for the localization branch, which could not well address OOD object detection that includes localization and classification. Different from the above works, in this paper, we propose a method of Two-Stream Information Bottleneck to reduce the impact of lacking unknown data and improve the discrimination ability via reversing the optimization objective of standard information bottleneck. Extensive experimental results demonstrate the effectiveness of our method.

**Information bottleneck.** Recent research Wang et al. (2022); Lee et al. (2021); Schulz et al. (2020) has shown that IB Tishby et al. (2000) is a promising mechanism to reveal the principle of neural networks through the lens of information stored in encoded representations of inputs. Given two random variables $X$ and $Y$, the optimization objective of IB can be described as follows:

$$\max_{T} I(T;Y) - \beta I(T;X), \tag{1}$$

where $I(T;X)$ and $I(T;Y)$ are the mutual information of representation $T$ towards inputs $X$ and labels $Y$, respectively. $\beta$ controls the tradeoff. By this optimization objective, the intermediate representation $T$ can be promoted to contain compact task-relevant information. Recently, some works Ahuja et al. (2021); Li et al. (2022) have shown that IB is helpful for extracting domain-invariant representations, which improves the generalization ability. Besides, there exist some works Schulz et al. (2020); Kim et al. (2021) that indicate using IB could capture object-related information, which is beneficial for boosting the performance and enhancing the interpretability. In this paper, we explore exploiting IB to obtain simulative OOD features for training and design a mixture of IB to further enhance the object-related information. Extensive experiments on OOD-OD and incremental object detection have shown that our method is instrumental in improving the discrimination ability.

## 3 TWO-STREAM INFORMATION BOTTLENECK

For unsupervised OOD-OD, the object detector is trained based on the ID data $\{X, Y, B\}$, where $X$ denotes the set of ID images, $Y$ is the label set, and $B$ indicates location information. During inference, give an image $x^*$ including OOD objects, the trained object detector should accurately distinguish ID objects (the output is 1) from OOD objects (the output is 0).

### 3.1 OBJECT-RELATED INFORMATION EXTRACTION

Since object detection involves two subtasks, i.e., object localization and classification, for OOD-OD, the model should first localize OOD objects and ID objects. Then, the model could accurately distinguish ID objects from OOD objects, while correctly classifying ID objects. To attain this goal,

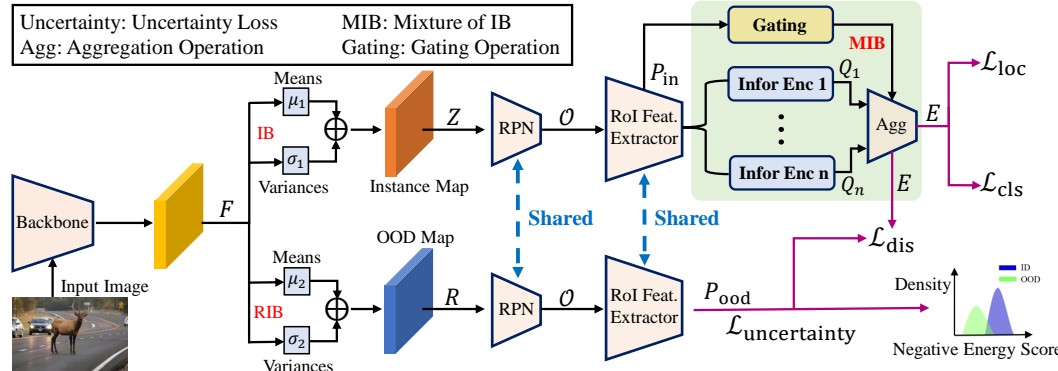

Figure 2: The architecture of our method. 'Infor Enc' indicates information enhancement. Our method mainly consists of an IB branch and a RIB branch. Particularly, the IB branch aims to capture plentiful object-related information by optimizing the objective of information bottleneck. Meanwhile, after obtaining the high-level features from the detection head, a mixture of IB is designed to further enhance object-related information, which is beneficial for improving the discrimination ability. Besides, to alleviate the impact of lacking unknown data for training, we propose a RIB to generate simulative OOD features by maximizing the loss $\mathcal{L}_{\mathrm{dis}}$. Here, it is worth noting that during the RPN process, for the OOD map, we only perform RoI Alignment based on the proposals $\mathcal{O}$ extracted from the Instance map. Finally, the ID features from the MIB and the simulative OOD features are all used to calculate the uncertainty loss $\mathcal{L}_{\mathrm{uncertainty}}$.

it is important to extract plentiful object-related information. As IB owns the advantage of capturing compact task-relevant information Alemi et al. (2017), we explore exploiting IB to compress object-irrelevant information (e.g., the background information) in the extracted features, which is beneficial for improving the discrimination performance.

Concretely, as shown in Fig. 2, we follow the baseline work Du et al. (2022b) and adopt the widely used object detector, i.e., Faster R-CNN Ren et al. (2015), as the basic detection model. Given an input image, we first employ the backbone network, e.g., ResNet He et al. (2016), to extract the corresponding feature map $F \in \mathbb{R}^{w \times h \times c}$, where $w$, $h$, and $c$ separately denote width, height, and the number of channels. To obtain rich object-related information, we exploit the constraint of variational information bottleneck Alemi et al. (2017) to further encode the feature map $F$. Specifically, we separately define a convolutional network $W_{\mu 1}$ and $W_{\sigma 1}$ to estimate the corresponding means and variances. The encoding processes of $F$ are shown as follows:

$$\mu_1 = W_{\mu 1} * F, \qquad \sigma_1 = W_{\sigma 1} * F, \qquad Z = \mu_1 + \epsilon \cdot \exp(\sigma_1), \tag{2}$$

where $\mu_1 \in \mathbb{R}^{w \times h \times c}$ and $\sigma_1 \in \mathbb{R}^{w \times h \times c}$ are the estimated means and variances. $\epsilon$ indicates Gaussian noise sampled from $\mathcal{N}(0, I)$. '$*$' represents the convolutional operation. The encoding output is denoted as the Instance map $Z \in \mathbb{R}^{w \times h \times c}$.

Next, $Z$ is taken as the input of the RPN module to extract a series of object proposals $\mathcal{O}$. Based on $\mathcal{O}$, RoI-Alignment operation followed by RoI-Feature extraction is performed on $Z$ to obtain the output $P_{\mathrm{in}} \in \mathbb{R}^{z \times s}$, where $z$ and $s$ respectively denote the number of proposals and channels. Since the object proposals usually contain much object-irrelevant information (e.g., the background information) that may weaken the discrimination performance, to this end, we design a mixture of information bottlenecks consisting of multiple branches to further enhance object-related information. For each branch, based on $P_{\mathrm{in}}$, we first define two fully-connected networks to separately estimate the corresponding means and variances. Then, we perform an encoding operation of $P_{\mathrm{in}}$:

$$P_i^\mu = \Phi_i^\mu(P_{\mathrm{in}}), \qquad P_i^\sigma = \Phi_i^\sigma(P_{\mathrm{in}}), \qquad Q_i = P_i^\mu + \epsilon \cdot \exp(P_i^\sigma), \tag{3}$$

where $i = 1, ..., n$. 'n' is the number of IB. $\Phi_i^\mu$ and $\Phi_i^\sigma$ represent two different fully-connected networks. $P_i^\mu \in \mathbb{R}^{z \times s}$ and $P_i^\sigma \in \mathbb{R}^{z \times s}$ are the estimated means and variances. $Q_i \in \mathbb{R}^{z \times s}$ denotes the output encoding results of the current branch.

Since $P_{\mathrm{in}}$ contains the information belonging to multiple different objects and much background information, exploiting multiple branches of information bottlenecks is beneficial for purifying object-related information from multiple different facets. Next, we first define a gating operation to aggregate the information from different IB. By means of the residual operation between the aggregated

information and the input $P_{\text{in}}$, the object-related information in $P_{\text{in}}$ could be enhanced. The overall enhancing processes are shown as follows:

$$G_i = \frac{\overline{P_{\text{in}}}^T \overline{Q_i}}{\sum_{i=1}^{n} \overline{P_{\text{in}}}^T \overline{Q_i}}, \qquad A = \sum_{i=1}^{n} G_i \cdot Q_i, \qquad E = A + \alpha \cdot P_{\text{in}}, \qquad (4)$$

where $\overline{P_{\text{in}}} \in \mathbb{R}^s$ and $\overline{Q_i} \in \mathbb{R}^s$ separately represent the average results of $P_{\text{in}}$ and $Q_i$. $G_i$ is the calculated gating weight. $A \in \mathbb{R}^{z \times s}$ indicates the aggregated results. Here, $\alpha \in \mathbb{R}^{z \times s}$ denotes the learnable sigmoid weight, i.e., $\alpha = \Psi(P_{\text{in}})$, where $\Psi$ is a fully-connected network. Finally, $E \in \mathbb{R}^{z \times s}$ indicates the output enhancing result.

As shown in Fig. 2, during training, $E$ is taken as the input of the classifier and regressor to calculate the classification and localization losses. The joint training objective is shown as follows:

$$\mathcal{L}_{\text{IB}} = \mathcal{L}_{\text{cls}} + \mathcal{L}_{\text{loc}} + \beta \cdot (\text{KL}[p(Z|F), r(Z)] + \frac{1}{n} \sum_{i=1}^{n} \text{KL}[p(Q_i|P_{\text{in}}), r(Q_i)]), \qquad (5)$$

where $\mathcal{L}_{\text{cls}}$ and $\mathcal{L}_{\text{loc}}$ separately denote the classification and localization losses. $\beta$ is a hyperparameter. In the experiments, $\beta$ is set to 0.0001. Following the information theories for deep learning Alemi et al. (2017), we define $r(\cdot)$ as a prior marginal distribution, which is modeled as a standard Gaussian $\mathcal{N}(0, I)$. Obviously, by minimizing the task loss and the $KL$-divergence loss, the dependence between $Z$ and $F$ and that between $Q_i$ and $P_{\text{in}}$ are reduced, indicating that $Z$ and $Q_i$ encode plentiful object-relevant information from the input $F$ and $P_{\text{in}}$, which is instrumental in improving the discrimination performance.

## 3.2 Simulative OOD Features Generation

For unsupervised OOD-OD, one of the major challenges lies in lacking unknown data for supervision, which is prone to producing overconfident incorrect predictions for OOD objects. To this end, we propose a RIB method to generate simulative OOD features by reversing the optimization objective of the standard IB Alemi et al. (2017), which reduces the impact of lacking unknown data and improves the ability of distinguishing OOD objects.

Concretely, as shown in Fig. 2, based on the feature map $F$ from the backbone network, we first define a convolutional network $W_{\mu 2}$ and $W_{\sigma 2}$ to estimate the corresponding means and variances and leverage variational information bottleneck Alemi et al. (2017) to encode $F$:

$$\mu_2 = W_{\mu 2} * F, \qquad \sigma_2 = W_{\sigma 2} * F, \qquad R = \mu_2 + \epsilon \cdot \exp(\sigma_2), \qquad (6)$$

where $R \in \mathbb{R}^{w \times h \times c}$ indicates the encoding OOD map. Next, based on the object proposals $\mathcal{O}$, RoI-Alignment operation followed by RoI-Feature extraction is performed on $R$ to obtain simulative OOD features $P_{\text{ood}} \in \mathbb{R}^{z \times s}$. To promote $P_{\text{ood}}$ to contain plentiful task-irrelevant information, we explore reversing the optimization objective of the standard IB (as shown in equation 1):

$$\max_{R, P_{\text{ood}}} I((R, P_{\text{ood}}); F) - \lambda I(P_{\text{ood}}; Y), \qquad (7)$$

where $\lambda$ controls the tradeoff. The goal of equation 7 is to enforce the extracted $R$ and $P_{\text{ood}}$ from the input $F$ to encode much less task-related information, which is beneficial for obtaining simulative OOD features for supervision. To attain this goal, we explore maximizing the prediction discrepancy $\mathcal{L}_{\text{dis}}$ between $E$ and $P_{\text{ood}}$. The processes are shown as follows:

$$\mathcal{L}_{\text{dis}} = \frac{1}{K} \sum_{k=1}^{K} |p(P_{\text{ood}})_k - p(E)_k|, \qquad (8)$$

where $p(P_{\text{ood}})_k$ and $p(E)_k$ denote the prediction probability for class $k$, respectively. $K$ is the number of ID categories. Meanwhile, the classifier with shared parameters is used to produce $p(P_{\text{ood}})$ and $p(E)$. Besides, since $R$ is directly encoded based on the input $F$, $R$ could be considered to be related to $F$. Here, we do not use a mutual information constraint to enhance the dependence between $R$ and $F$. By maximizing the loss $\mathcal{L}_{\text{dis}}$ while minimizing the task loss, the gap between $P_{\text{ood}}$ and $E$ will be enlarged, which promotes $P_{\text{ood}}$ to contain plentiful information unrelated to the ID objects. Finally, to achieve the goal of distinguishing OOD objects from ID objects, $P_{\text{ood}}$ and $E$

are used to calculate an uncertainty loss Du et al. (2022b), which aims to regularize the detector to produce a low OOD score for the ID object features, and a high OOD score for the simulative OOD features. The processes are shown as follows:

$$\mathcal{L}_{\text{uncertainty}} = \mathbb{E}_{u \backsim E}[-\log\frac{\exp^{-\mathcal{E}(u)}}{1 + \exp^{-\mathcal{E}(u)}}] + \mathbb{E}_{v \backsim P_{\text{ood}}}[-\log\frac{1}{1 + \exp^{-\mathcal{E}(v)}}], \tag{9}$$

where $\mathcal{E}(\cdot)$ is the object-level energy score Du et al. (2022b); Liu et al. (2020). During training, we can only access the ID data. The **overall training objective** is shown as follows:

$$\mathcal{L} = \mathcal{L}_{\text{IB}} - \lambda \cdot \mathcal{L}_{\text{dis}} + \tau \cdot \mathcal{L}_{\text{uncertainty}}, \tag{10}$$

where $\lambda$ and $\tau$ are two hyper-parameters, which are set to 0.001 and 0.1 in the experiments.

### 3.3 Inference for OOD Object Detection

During inference, we use the output of the uncertainty loss for OOD object detection. Specifically, for a predicted bounding box **b**, the detection processes are shown as follows:

$$score(\mathbf{b}) = \frac{\exp^{-\mathcal{E}(\mathbf{b})}}{1 + \exp^{-\mathcal{E}(\mathbf{b})}}, \qquad \mathcal{D}(\mathbf{b}) = \begin{cases} 0 & \text{if } score(\mathbf{b}) < \gamma, \\ 1 & \text{if } score(\mathbf{b}) \geq \gamma. \end{cases} \tag{11}$$

For the output of the classifier $\mathcal{D}(\cdot)$, the commonly used threshold mechanism is leveraged to distinguish the ID objects (the result is 1) from the OOD objects (the result is 0). The threshold $\gamma$ is set to 0.95 so that a high fraction of ID data is correctly classified.

## 4 Experiments

In the experiments, for unsupervised OOD-OD, we follow the settings of the work Du et al. (2022b) and do not use any auxiliary dataset for training. And our method is evaluated on two different benchmarks. Furthermore, to further demonstrate the effectiveness of our method, we verify our method on the task of class-incremental object detection Kj et al. (2021), i.e., new classes are sequentially introduced to the object detector.

### 4.1 Implementation Details and Benchmarks

**Implementation Details.** We use Faster R-CNN Ren et al. (2015) with the RoI-Alignment layer He et al. (2017) as the basic detection model. The backbone is ResNet-50 He et al. (2016). The parameters are pre-trained on ImageNet Russakovsky et al. (2015) for initialization. For the generation of the Instance map (equation 2) and OOD map (equation 6), we separately utilize two convolutional layers to define $W_{\mu 1}$, $W_{\sigma 1}$, $W_{\mu 2}$, and $W_{\sigma 2}$. For each branch of MIB (equation 3), we respectively utilize two fully-connected layers to define $\Phi^{\mu}$ and $\Phi^{\sigma}$. And the number n of the IB branches is set to 8. All the experiments are trained using the standard SGD optimizer with a learning rate of 0.02.

**OOD-OD Benchmarks.** PASCAL VOC Everingham et al. (2010) and Berkeley DeepDrive (BDD-100k) Yu et al. (2020) datasets are taken as the ID training data. Meanwhile, we adopt MS-COCO Lin et al. (2014) and OpenImages Kuznetsova et al. (2020) as the OOD datasets to evaluate the trained model. And the OOD datasets are manually examined to ensure the OOD images do not contain ID categories.

**Metrics.** To evaluate the performance of unsupervised OOD-OD, we report: (1) the false positive rate (FPR95) of OOD objects when the true positive rate of ID objects is at 95%; (2) the area under the receiver operating characteristic curve (AUROC); (3) mean average precision (mAP).

### 4.2 Performance Analysis of Unsupervised OOD-OD

Table 1 shows the performance of unsupervised OOD-OD. We can see that though these methods own similar detection performance, the ability of distinguishing OOD objects differs significantly. This shows that these detection methods are easily affected by OOD objects. Thus, detecting OOD objects is meaningful for promoting the safe application of object detectors. We can see that compared with baseline methods, our method obtains the best performance of OOD object detection.

Table 1: The performance (%) of unsupervised OOD-OD. All methods are trained based on ID data and do not use any auxiliary data. ↑ denotes larger values are better and ↓ denotes smaller values are better. '†' indicates that we directly run the released code to obtain the results. 'Bbone' and 'R50' separately represent backbone network and ResNet-50.

| ID Data | Method | Bbone (Params) | FPR95 ↓ | AUROC ↑ | mAP (ID)↑ |
|---|---|---|---|---|---|
| | | | OOD: MS-COCO / OpenImages | | |
| | MSP Hendrycks & Gimpel (2017) | - | 70.99 / 73.13 | 83.45 / 81.91 | 48.7 |
| | ODIN Liang et al. (2017) | - | 59.82 / 63.14 | 82.20 / 82.59 | 48.7 |
| | Mahalanobis (Lee et al., 2018b) | - | 67.73 / 65.41 | 81.45 / 81.48 | 48.7 |
| | Gram matrices Sastry & Oore (2020) | - | 62.75 / 67.42 | 79.88 / 77.62 | 48.7 |
| **PASCAL-** | Energy score Liu et al. (2020) | - | 56.89 / 58.69 | 83.69 / 82.98 | 48.7 |
| **VOC** | Generalized ODIN Hsu et al. (2020) | - | 59.57 / 70.28 | 83.12 / 79.23 | 48.1 |
| | CSI Tack et al. (2020) | - | 59.91 / 57.41 | 81.83 / 82.95 | 48.1 |
| | GAN-synthesis Lee et al. (2018a) | - | 60.93 / 59.97 | 83.67 / 82.67 | 48.5 |
| | †VOS (Baseline) Du et al. (2022b) | R50 (41.4M) | 51.97 / 56.81 | 87.55 / 83.37 | 48.1 |
| | **Two-Stream IB** | R50 (54.7M) | **41.55 / 47.19** | **90.36 / 88.09** | **49.2** |
| | MSP Hendrycks & Gimpel (2017) | - | 80.94 / 79.04 | 75.87 / 77.38 | 31.2 |
| | ODIN Liang et al. (2017) | - | 62.85 / 58.92 | 74.44 / 76.61 | 31.2 |
| | Mahalanobis Lee et al. (2018b) | - | 55.74 / 47.69 | 85.71 / 88.05 | 31.2 |
| | Gram matrices Sastry & Oore (2020) | - | 60.93 / 77.55 | 74.93 / 59.38 | 31.2 |
| **BDD-** | Energy score Liu et al. (2020) | - | 60.06 / 54.97 | 77.48 / 79.60 | 31.2 |
| **100k** | Generalized ODIN Hsu et al. (2020) | - | 57.27 / 50.17 | 85.22 / 87.18 | **31.8** |
| | CSI Tack et al. (2020) | - | 47.10 / 37.06 | 84.09 / 87.99 | 30.6 |
| | GAN-synthesis Lee et al. (2018a) | - | 57.03 / 50.61 | 78.82 / 81.25 | 31.4 |
| | †VOS (Baseline) Du et al. (2022b) | R50 (41.4M) | 50.25 / 41.06 | 83.92 / 86.80 | 31.1 |
| | **Two-Stream IB** | R50 (54.7M) | **36.85 / 24.00** | **88.47 / 92.54** | 31.3 |

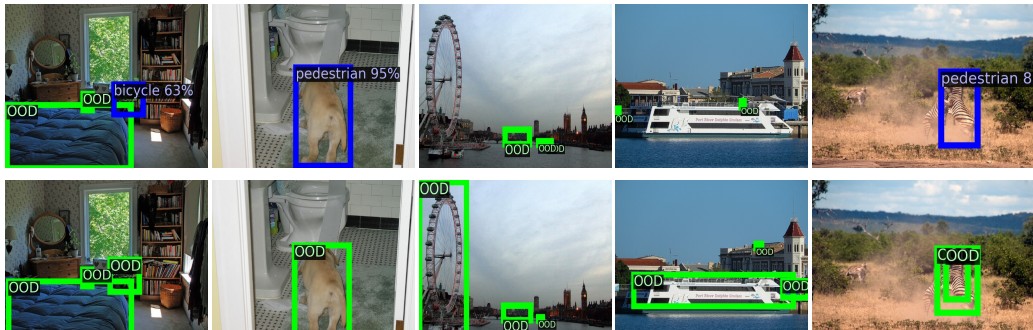

Figure 3: Detection results on the OOD images from MS-COCO. The first and second rows respectively indicate results based on VOS Du et al. (2022b) and our method. The in-distribution dataset is BDD-100k. Blue boxes represent objects detected and classified as one of the ID categories. Green boxes indicate OOD objects. We can see that our method accurately determines OOD objects.

Particularly, compared with VOS Du et al. (2022b) that aims to synthesize virtual outliers, based on FPR95, for PASCAL VOC Everingham et al. (2010), our method outperforms VOS by around 10.42% and 9.62%. For BDD-100k Yu et al. (2020), our method outperforms VOS by around 13.4% and 17.06%. This shows that our method of Two-Stream Information Bottleneck is beneficial for extracting simulative OOD features, which reduces the impact of lacking unknown data for supervision and improves the discrimination ability of the object detector.

In Fig. 3, we show some detection examples. For these images, the baseline method Du et al. (2022b) does not detect all OOD objects accurately. Taking the second column as an example, VOS Du et al. (2022b) misclassifies the dog into the Pedestrian category. We can see that our method correctly localizes and recognizes OOD objects, further demonstrating its effectiveness.

### 4.3 PERFORMANCE ANALYSIS OF CLASS-INCREMENTAL OBJECT DETECTION

To further demonstrate the effectiveness of our method, we evaluate our method on class-incremental object detection Kj et al. (2021) and follow the standard evaluation protocol Kj et al. (2021). We initially learn 10, 15, or 19 base classes, and then introduce 10, 5, or 1 new class as the second task. Meanwhile, we directly plug our method into the baseline method Kj et al. (2021) and do not calculate the uncertainty loss (equation 9). The training details are the same as the baseline Kj et al.

Table 2: Performance (%) analysis of class-incremental object detection based on PASCAL VOC Everingham et al. (2010). We consider adding 10, 5, and 1 classes (highlighted in blue) to a detector trained on the rest of the classes. 'iOD + Ours' indicates that our method is plugged into iOD Kj et al. (2021). Here, 'P50' indicates that the mAP metric is calculated when the IOU threshold is set to 0.5.

| 10 + 10 setting | aero | cycle | bird | boat | bottle | bus | car | cat | chair | cow | table | dog | horse | bike | person | plant | sheep | sofa | train | tv | P50 |
|---|---|---|---|---|---|---|---|---|---|---|---|---|---|---|---|---|---|---|---|---|---|
| All 20 | 79.4 | 83.3 | 73.2 | 59.4 | 62.6 | 81.7 | 86.6 | 83 | 56.4 | 81.6 | 71.9 | 83 | 85.4 | 81.5 | 82.7 | 49.4 | 74.4 | 75.1 | 79.6 | 73.6 | 75.2 |
| First 10 | 78.6 | 78.6 | 72 | 54.5 | 63.9 | 81.5 | 87 | 78.2 | 55.3 | 84.4 | - | - | - | - | - | - | - | - | - | - | 73.4 |
| Std Training | 35.7 | 9.1 | 16.6 | 7.3 | 9.1 | 18.2 | 9.1 | 26.4 | 9.1 | 6.1 | 57.6 | 57.1 | 72.6 | 67.5 | 73.9 | 33.5 | 53.4 | 61.1 | 66.5 | 57 | 37.3 |
| Shmelkov et al. Shmelkov et al. (2017) | 69.9 | 70.4 | 69.4 | 54.3 | 48 | 68.7 | 78.9 | 68.4 | 45.5 | 58.1 | 59.7 | 72.7 | 73.5 | 73.2 | 66.3 | 29.5 | 63.4 | 61.6 | 69.3 | 62.2 | 63.1 |
| Faster ILOD Peng et al. (2020) | 72.8 | 75.7 | 71.2 | 60.5 | 61.7 | 70.4 | 83.3 | 76.6 | 53.1 | 72.3 | 36.7 | 70.9 | 66.8 | 67.6 | 66.1 | 24.7 | 63.1 | 48.1 | 57.1 | 43.6 | 62.2 |
| ORE Joseph et al. (2021) | 63.5 | 70.9 | 58.9 | 42.9 | 34.1 | 76.2 | 80.7 | 76.3 | 34.1 | 66.1 | 56.1 | 70.4 | 80.2 | 72.3 | 81.8 | 42.7 | 71.6 | 68.1 | 77 | 67.7 | 64.6 |
| OW-DETR Gupta et al. (2022) | 61.8 | 69.1 | 67.8 | 45.8 | 47.3 | 78.3 | 78.4 | 78.6 | 36.2 | 71.5 | 57.5 | 75.3 | 76.2 | 77.4 | 79.5 | 40.1 | 66.8 | 66.3 | 75.6 | 64.1 | 65.7 |
| ROSETTA Yang et al. (2022) | 74.2 | 76.2 | 64.9 | 54.4 | 57.4 | 76.1 | 84.4 | 68.8 | 52.4 | 67.0 | 62.9 | 63.3 | 79.8 | 72.8 | 78.1 | 40.1 | 62.3 | 61.2 | 72.4 | 66.8 | 66.8 |
| iOD Kj et al. (2021) | 76 | 74.6 | 67.5 | 55.9 | 57.6 | 75.1 | 85.4 | 77 | 43.7 | 70.8 | 60.1 | 66.4 | 76 | 72.6 | 74.6 | 39.7 | 64 | 60.2 | 68.5 | 60.5 | 66.3 |
| iOD + Ours | 77.6 | 76.5 | 71.5 | 57.4 | 56.4 | 80.6 | 85.8 | 80.4 | 48.4 | 77.1 | 54.9 | 75.4 | 79.4 | 75.8 | 79.4 | 45.5 | 73.8 | 68.5 | 73.0 | 69.4 | **70.4** |
| 15 + 5 setting | aero | cycle | bird | boat | bottle | bus | car | cat | chair | cow | table | dog | horse | bike | person | plant | sheep | sofa | train | tv | P50 |
| All 20 | 79.4 | 83.3 | 73.2 | 59.4 | 62.6 | 81.7 | 86.6 | 83 | 56.4 | 81.6 | 71.9 | 83 | 85.4 | 81.5 | 82.7 | 49.4 | 74.4 | 75.1 | 79.6 | 73.6 | 75.2 |
| First 15 | 78.1 | 82.6 | 74.2 | 61.8 | 63.9 | 80.4 | 87 | 81.5 | 57.7 | 80.4 | 73.1 | 80.8 | 85.8 | 81.6 | 83.9 | - | - | - | - | - | 53.2 |
| Std Training | 12.7 | 0.6 | 9.1 | 9.1 | 3 | 0 | 8.5 | 9.1 | 0 | 3 | 9.1 | 0 | 3.3 | 2.3 | 9.1 | 37.6 | 51.2 | 57.8 | 51.5 | 59.8 | 16.8 |
| Shmelkov et al. Shmelkov et al. (2017) | 70.5 | 79.2 | 68.8 | 59.1 | 53.2 | 75.4 | 79.4 | 78.8 | 46.6 | 59.4 | 59 | 75.8 | 71.8 | 78.6 | 69.6 | 33.7 | 61.5 | 63.1 | 71.7 | 62.2 | 65.9 |
| Faster ILOD Peng et al. (2020) | 66.5 | 78.1 | 71.8 | 54.6 | 61.4 | 68.4 | 82.6 | 82.7 | 52.1 | 74.3 | 63.1 | 78.6 | 80.5 | 78.4 | 80.4 | 36.7 | 61.7 | 59.3 | 67.9 | 59.1 | 67.9 |
| ORE Joseph et al. (2021) | 75.4 | 81 | 67.1 | 51.9 | 55.7 | 77.2 | 85.6 | 81.7 | 46.1 | 76.2 | 55.4 | 76.7 | 86.2 | 78.5 | 82.1 | 32.8 | 63.6 | 54.7 | 77.7 | 64.6 | 68.5 |
| OW-DETR Gupta et al. (2022) | 77.1 | 76.5 | 69.2 | 51.3 | 61.3 | 79.8 | 84.2 | 81.0 | 49.7 | 79.6 | 58.1 | 79.0 | 83.1 | 67.8 | 85.4 | 33.2 | 65.1 | 62.0 | 73.9 | 65.0 | 69.4 |
| ROSETTA Yang et al. (2022) | 76.5 | 77.5 | 65.1 | 56.0 | 60.0 | 78.3 | 85.5 | 78.7 | 49.5 | 68.2 | 67.4 | 71.2 | 83.9 | 75.7 | 82.0 | 43.0 | 60.6 | 64.1 | 72.8 | 67.4 | 69.2 |
| iOD Kj et al. (2021) | 78.4 | 79.7 | 66.9 | 54.8 | 56.2 | 77.7 | 84.6 | 79.1 | 47.7 | 75 | 64.8 | 74.7 | 81.6 | 77.5 | 80.2 | 37.8 | 58 | 54.6 | 73 | 56.1 | 67.8 |
| iOD + Ours | 78.0 | 78.7 | 73.6 | 53.8 | 63.6 | 80.1 | 85.6 | 83.1 | 50.8 | 80.9 | 66.5 | 81.0 | 83.1 | 77.2 | 77.7 | 39.8 | 66.6 | 63.0 | 71.8 | 66.0 | **71.1** |
| 19 + 1 setting | aero | cycle | bird | boat | bottle | bus | car | cat | chair | cow | table | dog | horse | bike | person | plant | sheep | sofa | train | tv | P50 |
| All 20 | 79.4 | 83.3 | 73.2 | 59.4 | 62.6 | 81.7 | 86.6 | 83 | 56.4 | 81.6 | 71.9 | 83 | 85.4 | 81.5 | 82.7 | 49.4 | 74.4 | 75.1 | 79.6 | 73.6 | 75.2 |
| First 19 | 76.3 | 77.3 | 68.4 | 55.4 | 59.7 | 81.4 | 85.3 | 80.3 | 47.8 | 78.1 | 65.7 | 77.5 | 83.5 | 76.2 | 77.2 | 46.6 | 71.4 | 65.8 | 76.5 | - | 67.5 |
| Std Training | 16.6 | 9.1 | 9.1 | 9.1 | 9.1 | 8.3 | 35.3 | 9.1 | 0 | 22.3 | 9.1 | 9.1 | 9.1 | 13.7 | 9.1 | 9.1 | 23.1 | 9.1 | 15.4 | 50.7 | 14.3 |
| Shmelkov et al. Shmelkov et al. (2017) | 69.4 | 79.3 | 69.5 | 57.4 | 45.4 | 78.4 | 79.1 | 80.5 | 45.7 | 76.3 | 64.8 | 77.2 | 80.8 | 77.5 | 70.1 | 42.3 | 67.5 | 64.4 | 76.7 | 62.7 | 68.3 |
| Faster ILOD Peng et al. (2020) | 64.2 | 74.7 | 73.2 | 55.5 | 53.7 | 70.8 | 82.9 | 82.6 | 51.6 | 79.7 | 58.7 | 78.8 | 81.8 | 75.3 | 77.4 | 43.1 | 73.8 | 61.7 | 69.8 | 61.1 | 68.6 |
| ORE Joseph et al. (2021) | 67.3 | 76.8 | 60 | 48.4 | 58.8 | 81.1 | 86.5 | 75.8 | 41.5 | 79.6 | 54.6 | 72.8 | 85.9 | 81.7 | 82.4 | 44.8 | 75.8 | 68.2 | 75.7 | 60.1 | 68.9 |
| OW-DETR Gupta et al. (2022) | 70.5 | 77.2 | 73.8 | 54.0 | 55.6 | 79.0 | 80.8 | 80.6 | 43.2 | 80.4 | 53.5 | 77.5 | 89.5 | 82.0 | 74.7 | 43.3 | 71.9 | 66.6 | 79.4 | 62.0 | 70.2 |
| ROSETTA Yang et al. (2022) | 75.3 | 77.9 | 65.3 | 56.2 | 55.3 | 79.6 | 84.6 | 79.2 | 49.2 | 73.7 | 68.3 | 71.0 | 78.9 | 77.7 | 80.7 | 44.0 | 69.6 | 68.5 | 76.1 | 68.3 | 69.6 |
| iOD Kj et al. (2021) | 78.2 | 77.5 | 69.4 | 55 | 56 | 78.4 | 84.2 | 79.2 | 46.6 | 79 | 63.2 | 78.5 | 82.7 | 79.1 | 79.9 | 44.1 | 73.2 | 66.3 | 76.4 | 57.6 | 70.2 |
| iOD + Ours | 77.4 | 78.3 | 73.6 | 58.2 | 62.3 | 77.5 | 85.2 | 81.4 | 51.5 | 78.4 | 62.3 | 81.9 | 84.2 | 74.7 | 77.2 | 49.7 | 75.6 | 67.6 | 74.3 | 57.8 | **71.5** |

Table 3: Performance (%) analysis of incremental object detection based on PASCAL VOC Everingham et al. (2010). Here, the IOU threshold is set to 0.75.

| 10 + 10 setting | aero | cycle | bird | boat | bottle | bus | car | cat | chair | cow | table | dog | horse | bike | person | plant | sheep | sofa | train | tv | P75 |
|---|---|---|---|---|---|---|---|---|---|---|---|---|---|---|---|---|---|---|---|---|---|
| iOD Kj et al. (2021) | 39.0 | 36.5 | 28.4 | 19.4 | 24.2 | 47.2 | 56.7 | 41.0 | 19.1 | 48.0 | 21.1 | 32.1 | 43.0 | 36.3 | 40.0 | 14.8 | 40.1 | 36.5 | 37.3 | 45.3 | 35.3 |
| iOD + Ours | 46.0 | 44.7 | 31.4 | 25.8 | 26.0 | 62.9 | 61.8 | 45.0 | 22.4 | 46.6 | 19.3 | 33.7 | 37.1 | 39.8 | 39.3 | 15.8 | 47.5 | 34.1 | 40.0 | 45.0 | **38.2** |
| 15 + 5 setting | aero | cycle | bird | boat | bottle | bus | car | cat | chair | cow | table | dog | horse | bike | person | plant | sheep | sofa | train | tv | P75 |
| iOD Kj et al. (2021) | 40.7 | 40.9 | 28.7 | 19.1 | 23.8 | 61.6 | 56.1 | 38.8 | 23.6 | 47.5 | 18.7 | 40.1 | 40.2 | 41.5 | 39.8 | 9.1 | 40.6 | 32.4 | 41.9 | 47.6 | 36.6 |
| iOD + Ours | 43.8 | 44.2 | 34.6 | 20.1 | 32.1 | 56.2 | 62.7 | 45.4 | 25.3 | 49.0 | 30.0 | 42.2 | 50.4 | 44.5 | 42.3 | 11.6 | 44.2 | 36.0 | 48.0 | 42.3 | **40.3** |
| 19 + 1 setting | aero | cycle | bird | boat | bottle | bus | car | cat | chair | cow | table | dog | horse | bike | person | plant | sheep | sofa | train | tv | P75 |
| iOD Kj et al. (2021) | 35.9 | 44.7 | 31.6 | 22.4 | 26.9 | 52.0 | 56.5 | 38.7 | 21.6 | 48.4 | 21.2 | 35.9 | 37.9 | 30.7 | 38.7 | 17.2 | 38.5 | 34.2 | 40.7 | 46.6 | 36.0 |
| iOD + Ours | 43.5 | 46.6 | 36.1 | 22.0 | 32.0 | 60.3 | 63.4 | 45.2 | 24.4 | 48.2 | 30.5 | 41.9 | 50.1 | 41.5 | 42.2 | 17.5 | 43.7 | 36.8 | 49.6 | 47.5 | **41.2** |

(2021). Table 2 and 3 separately show the results based on the metric of mAP50 and mAP75. We can see that plugging our method improves the detection performance significantly. Particularly, for the '19+1' setting and the mAP75 metric, employing our method boosts the performance by around 5.2%, which further shows that our method could indeed enhance the discrimination ability.

## 4.4 ABLATION AND VISUALIZATION ANALYSIS

In this section, we utilize PASCAL VOC as the ID data for training and MS-COCO as the OOD data to perform an ablation analysis of our method.

**Analysis of IB and RIB.** In this paper, we define an IB branch and a RIB branch to separately extract instance-level features and simulative OOD features. Here, we make an ablation analysis of our method. Table 4 shows the performance. Here, 'IB' includes the designed mixture of information bottlenecks and uses

Table 4: The performance (%) of only using the IB branch and only using the RIB branch.

| Method | FPR95 ↓ | AUROC ↑ | mAP (ID)↑ |
|---|---|---|---|
| VOS | 51.97 | 87.55 | 48.1 |
| IB | 46.71 | 89.17 | 48.9 |
| RIB | 45.14 | 89.94 | 49.1 |
| TIB | **41.55** | **90.36** | **49.2** |

VOS Du et al. (2022b) to synthesize virtual outliers for training. We can see that employing the IB branch could improve the detection performance, which shows that using the IB is indeed helpful for compressing object-unrelated contents. And we observe that exploiting the mixture of information

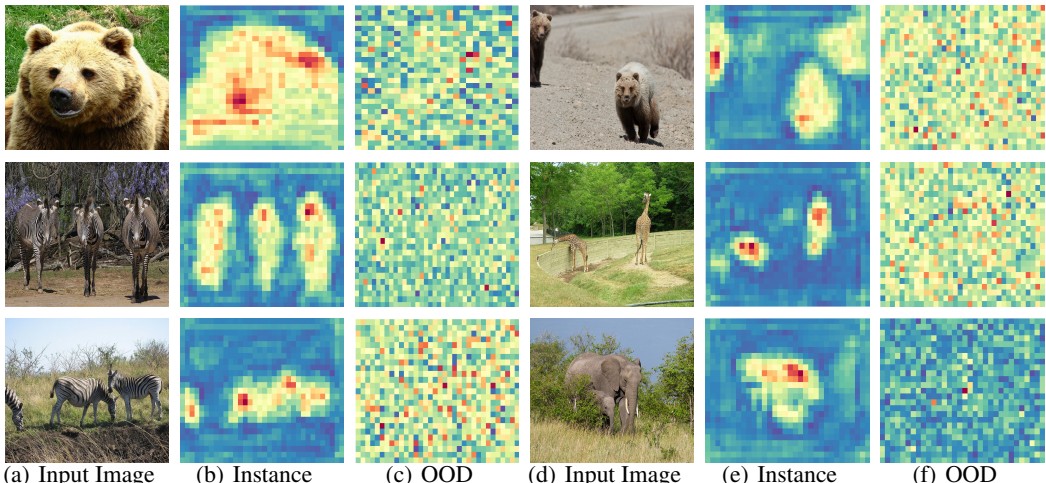

| (a) Input Image | (b) Instance | (c) OOD | (d) Input Image | (e) Instance | (f) OOD |

Figure 4: Visualization of the Instance map and OOD map based on the OOD data (MS-COCO). For each feature map, the channels corresponding to the maximum value are selected for visualization.

bottlenecks reduces FPR95 by around 2.8%, indicating that this module is beneficial for enhancing object-related information and improving the discrimination ability. Furthermore, we observe that only using the RIB module could obtain superior performance compared with the baseline method Du et al. (2022b). This indicates that reversing the optimization objective of the information bottleneck could extract object-unrelated information, which is helpful for obtaining simulative OOD features and improving the discrimination ability.

**Analysis of the branch number in MIB.** In this paper, we design a mixture of information bottlenecks (MIB) to purify object-related information from multiple different facets (as shown in Fig. 2). We make an ablation analysis of the branch number. Here, we only change the branch number and keep other modules unchanged. Table 5 shows the detection results. We can see that the number of IB branches indeed affects the performance of OOD detection. When the number of IB branches is small, the MIB module does not obtain

Table 5: The performance (%) of using a different number of IB branches.

| Num (n) | FPR95 ↓ | AUROC ↑ | mAP (ID)↑ |
|---------|---------|---------|-----------|
| 2 | 43.64 | 90.02 | **49.3** |
| 4 | 42.33 | **90.37** | 48.8 |
| 8 | **41.55** | 90.36 | 49.2 |
| 12 | 42.53 | 90.04 | 49.0 |

better performance. The reason may be that the proposal features involve much information belonging to different objects. Using fewer IB branches could not sufficiently capture object-related information, which weakens the discrimination. Instead, using more IB branches introduces more parameters, which may lead to overfitting and reduces the performance of OOD detection. We observe that for our method, the performance of using 8 branches is the best.

**Visualization analysis.** In this paper, we separately extract an Instance map and an OOD map based on the backbone output. In Fig. 4, we make a visualization analysis. We can see that the Instance map contains plentiful object-related contents and less object-irrelevant information, which is instrumental in improving the localization and recognition performance. Meanwhile, we observe the extracted OOD map is significantly unrelated to the current object, which is beneficial for obtaining simulative OOD features and alleviates the impact of lacking unknown data for supervision.

## 5 CONCLUSION

For unsupervised OOD-OD, this paper proposes a method of Two-Stream Information Bottleneck consisting of an IB branch and a RIB branch. Specifically, the IB branch aims to extract object-related information that is helpful for improving localization and recognition performance. Meanwhile, the RIB branch is to extract simulative OOD features to alleviate the impact of lacking unknown data for training by reversing the optimization objective of the information bottleneck. Extensive experimental results on OOD-OD and class-incremental object detection, and visualization analysis indicate the superiorities of our method.

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

# A  APPENDIX

Here we provide additional analyses, various ablation studies, and more visualization results.

## A.1  FURTHER DISCUSSION ABOUT RIB

In this paper, to alleviate the impact of lacking unknown data for supervision, we design a RIB module to extract simulative OOD features via reversing the optimization objective of the information bottleneck. The reversed objective is shown as follows:

$$\max_{R, P_{\text{ood}}} I((R, P_{\text{ood}}); F) - \lambda I(P_{\text{ood}}; Y). \tag{12}$$

**Proposition 1.** Assume $I(P_{\text{ood}}; Y) = 0$, then we achieve the features that contain plentiful out-of-distribution information.

*Proof.* Note that $I(P_{\text{ood}}; Y) = 0$ implies $P_{\text{ood}}$ and the labels $Y$ are independent. By minimizing the classification loss, the in-distribution features $P_{\text{in}}$ can be promoted to be related to the labels $Y$. Hence, $I(P_{\text{ood}}; Y) = 0$ enforces $P_{\text{ood}}$ to be irrelevant to $P_{\text{in}}$, which promotes $P_{\text{ood}}$ to contain rich out-of-distribution information.

## A.2  TRAINING AND INFERENCE PROCESSES

---
**Algorithm 1** Two-Stream Information Bottleneck for Unsupervised OOD-OD

---
**Input:** ID data $\{X, Y, B\}$, randomly initialized detector with parameter $\theta$, weight $\beta$ for the $KL$-loss, weight $\lambda$ for the loss $\mathcal{L}_{\text{dis}}$, weight $\tau$ for the uncertainty loss $\mathcal{L}_{\text{uncertainty}}$.
**Output:** Object detector with parameter $\theta^*$, and OOD detector $\mathcal{D}$.
**while** *train* **do**
  Sample images from the ID dataset $\{X, Y, B\}$.
  Calculate the Instance map $Z$ and OOD map $R$ using equation 2 and equation 6.
  Calculate the enhancing result $E$ using equation 3 and equation 4.
  Calculate the loss $\mathcal{L}_{\text{IB}}$ using equation 5, the loss $\mathcal{L}_{\text{dis}}$ using equation 8, the uncertainty loss $\mathcal{L}_{\text{uncertainty}}$ using equation 9.
  Update the parameters $\theta$ based on equation 10.
**end**
**while** *eval* **do**
  Calculate the OOD uncertainty score using the left part of equation 11.
  Perform thresholding comparison using the right part of equation 11.
**end**

---

## A.3  ABLATION ANALYSIS OF HYPER-PARAMETERS

In this paper, we use the hyper-parameter $\beta$ to control the $KL$-loss (equation 5) and use $\lambda$ and $\tau$ to separately control the loss $\mathcal{L}_{\text{dis}}$ and $\mathcal{L}_{\text{uncertainty}}$ (equation 10). Since $\mathcal{L}_{\text{cls}}$, $\mathcal{L}_{\text{loc}}$, and $\mathcal{L}_{\text{uncertainty}}$ are directly related to the task, $\beta$ and $\lambda$ should be set to a smaller value than $\tau$. Meanwhile, if $\beta$ and $\lambda$ are set to a very small value, $KL$-loss and $\mathcal{L}_{\text{dis}}$ may play a small role in optimization. Thus, it is meaningful to set proper values for these hyper-parameters. Next, we utilize PASCAL VOC as the ID data and MS-COCO as the OOD data to perform an ablation analysis of hyper-parameters. And we only change the value of hyper-parameters and keep other modules unchanged.

**Analysis of hyper-parameter $\beta$.** We use $\beta$ to control the $KL$-divergence loss in information bottlenecks. When $\beta$ is separately set to 0.001, 0.0001, and 0.00001, for FPR95, the corresponding performance is 43.84%, 41.55%, and 42.83%.

**Analysis of hyper-parameter $\lambda$.** We use $\lambda$ to constrain the reversing optimization objective of the information bottleneck. When $\lambda$ is separately set to 0.01, 0.001, and 0.0001, for FPR95, the corresponding performance is 44.15%, 41.55%, and 43.26%.

**Analysis of hyper-parameter $\tau$.** For the uncertainty loss, we follow the work Du et al. (2022b) and use the same setting for $\tau$. When $\tau$ is separately set to 0.15, 0.1, and 0.05, the corresponding performance is 44.58%, 41.55%, and 43.73%.

## A.4 More Experimental Results

**Results on RegNetX-4.0GF.** To further verify the effectiveness of our method, we evaluate our method on another backbone network, i.e., RegNetX-4.0GF Radosavovic et al. (2020). Here, we take PASCAL VOC as the ID training data and MS-COCO as the OOD data for evaluation. Based on FPR95, AUROC, and mAP, the performance of our method is 44.90%, 90.08%, and 51.5%, which significantly outperforms the VOS's performance, i.e., 50.81%, 88.42%, and 50.8%. This shows that our method of Two-Stream Information Bottleneck is able to strengthen the instance-related information and extract proper simulative OOD features, which reduces the impact of lacking unknown data for supervision and improves the discrimination ability.

**Further Analysis about RIB.** For RIB (see equation 7), since the OOD map $R$ is extracted based on the input feature map $F$, $R$ could be considered to be related to $F$. Hence, we do not use a mutual information constraint to enhance the dependence between $R$ and $F$. Here, we make an ablation analysis of adding the mutual information constraint. We take PASCAL VOC as the ID training data and MS-COCO as the OOD data for evaluation. We observe that adding the mutual information constraint increases the performance of FPR95 by around 1.1%. Meanwhile, we replace the variational operation with the traditional convolution operation. The performance of FPR95 is increased by around 4.8%. These analyses show that using the variational operation is helpful for capturing distribution-related information, which is better to extract out-of-distribution information.

## A.5 More Visualization Examples

In Fig. 5, we give more visualization examples. We can see that the extracted Instance map contains plentiful object-relevant information, which is helpful for localizing and recognizing objects accurately. Meanwhile, we can also see that the calculated OOD map is significantly different from the Instance map, which conforms to the meaning of the OOD, i.e., the OOD data deviates from the in-distribution. This further shows that our proposed reverse information bottleneck could indeed extract simulative OOD features, which is beneficial for alleviating the impact of lacking unknown data for supervision and improving the discrimination ability.

In Fig. 6 and 7, we show more detection results. We can see that our method accurately localizes and recognizes ID objects and OOD objects. Particularly, compared with ID objects that contain a fixed number of categories, the category distribution of OOD objects is diverse, presenting a significant challenge for the object detector. Our method attempts to solve this problem from the feature perspective, which has been demonstrated to be effective.

## A.6 Computation of Mutual Information on 3D Feature Maps

In this paper, we compute $KL$-divergence loss to approximate the mutual information. The processes are shown as follows:

$$\mathrm{KL}(p||q) = \sum p(x) \log \frac{p(x)}{q(x)},$$

(13)

where $p(\cdot)$ and $q(\cdot)$ represent two probability distributions. For example, given two 3D feature maps $H \in \mathbb{R}^{w \times h \times c}$ and $C \in \mathbb{R}^{w \times h \times c}$, we first perform softmax operation on $H$ and $C$. Then, we separately take the corresponding elements between the two processed results as the input of equation 13 to calculate the $KL$ result. The mean of the $KL$ results from all corresponding elements is taken as the output $KL$-divergence loss.

## A.7 Performance on Higher Levels of Contamination from OOD Classes

We select 8K images from COCO and synthesize some OOD objects on these images to perform a further evaluation. In Fig. 8, we show some synthesized images. Through experiments, we observe that our method still outperforms VOS Du et al. (2022b) significantly. Particularly, compared with VOS Du et al. (2022b), our method reduces FPR95 by around 10.1% and improves AUROC by around 3.2%, which further demonstrates the effectiveness of our method.

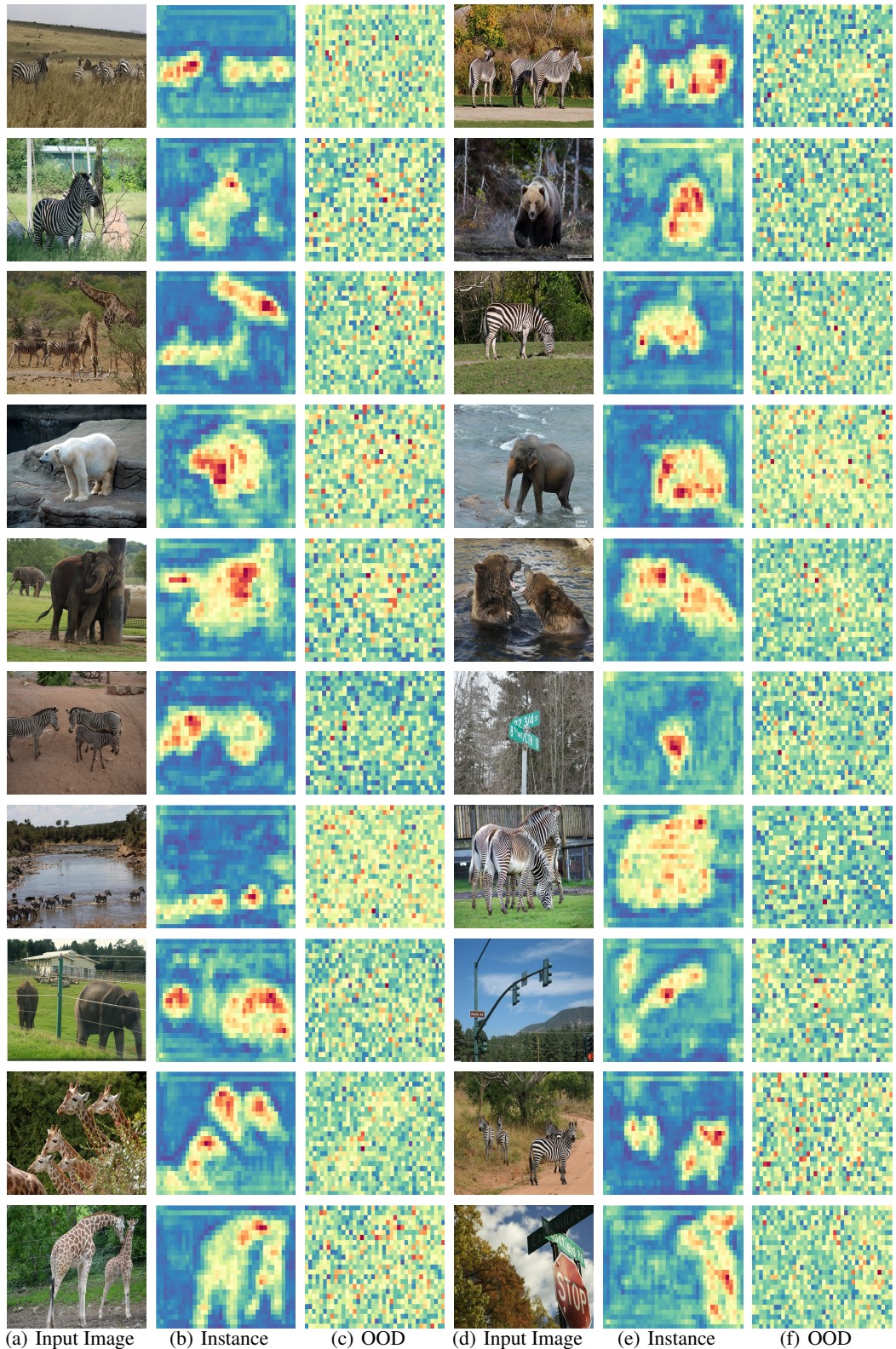

(a) Input Image    (b) Instance    (c) OOD    (d) Input Image    (e) Instance    (f) OOD

Figure 5: Visualization of the Instance map and OOD map. For each feature map, the channels corresponding to the maximum value are selected for visualization. We can see that the Instance map contains plentiful object-relevant information. Meanwhile, the OOD map involves sufficient object-irrelevant information, which is helpful for extracting simulative OOD features to alleviate the impact of lacking unknown data for supervision.

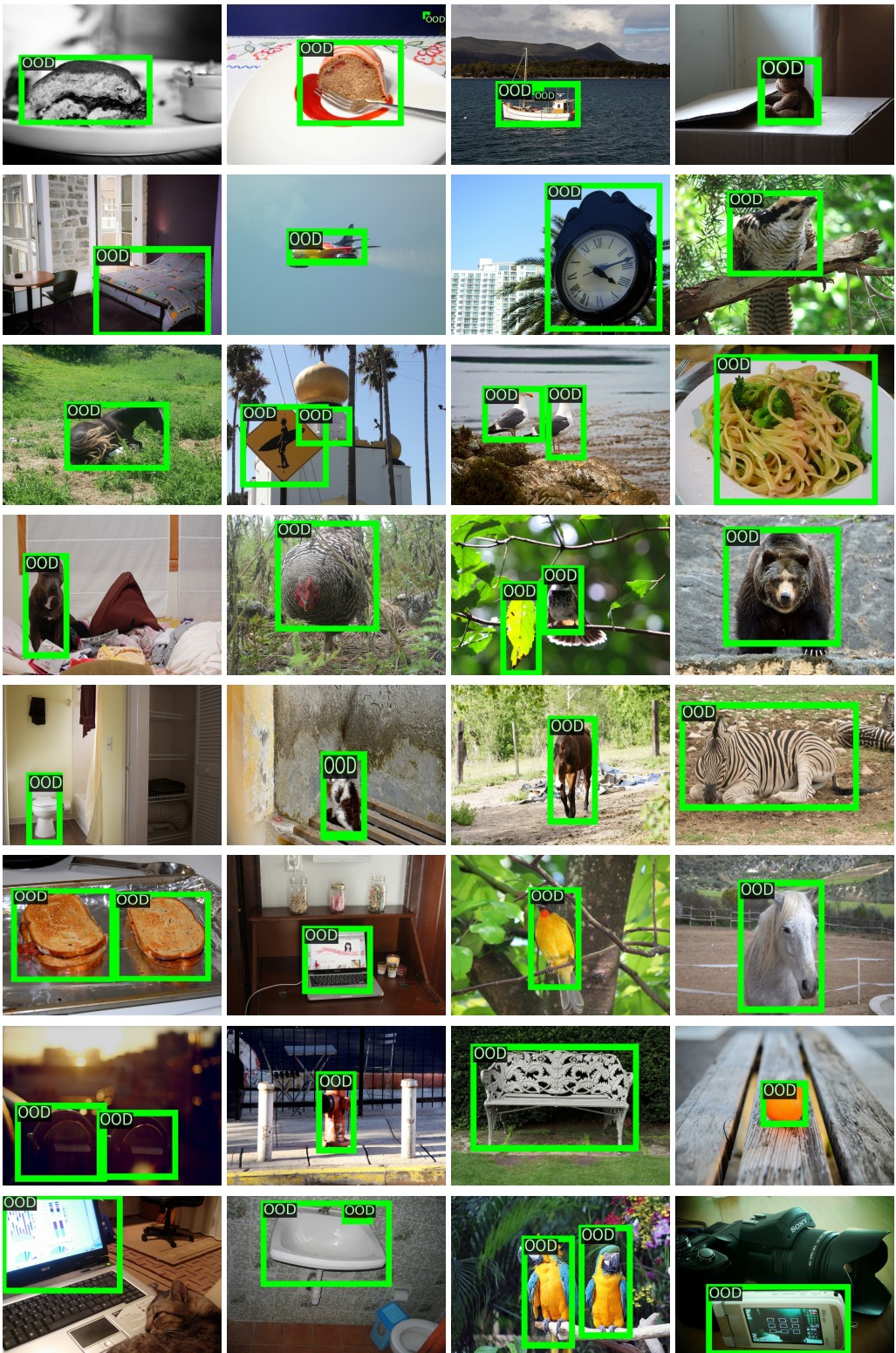

Figure 6: OOD detection examples based on our method. Here, we use BDD-100k as the in-distribution data and MS-COCO as the OOD data. We can see that our method accurately distinguishes OOD objects, which shows the effectiveness of our method.

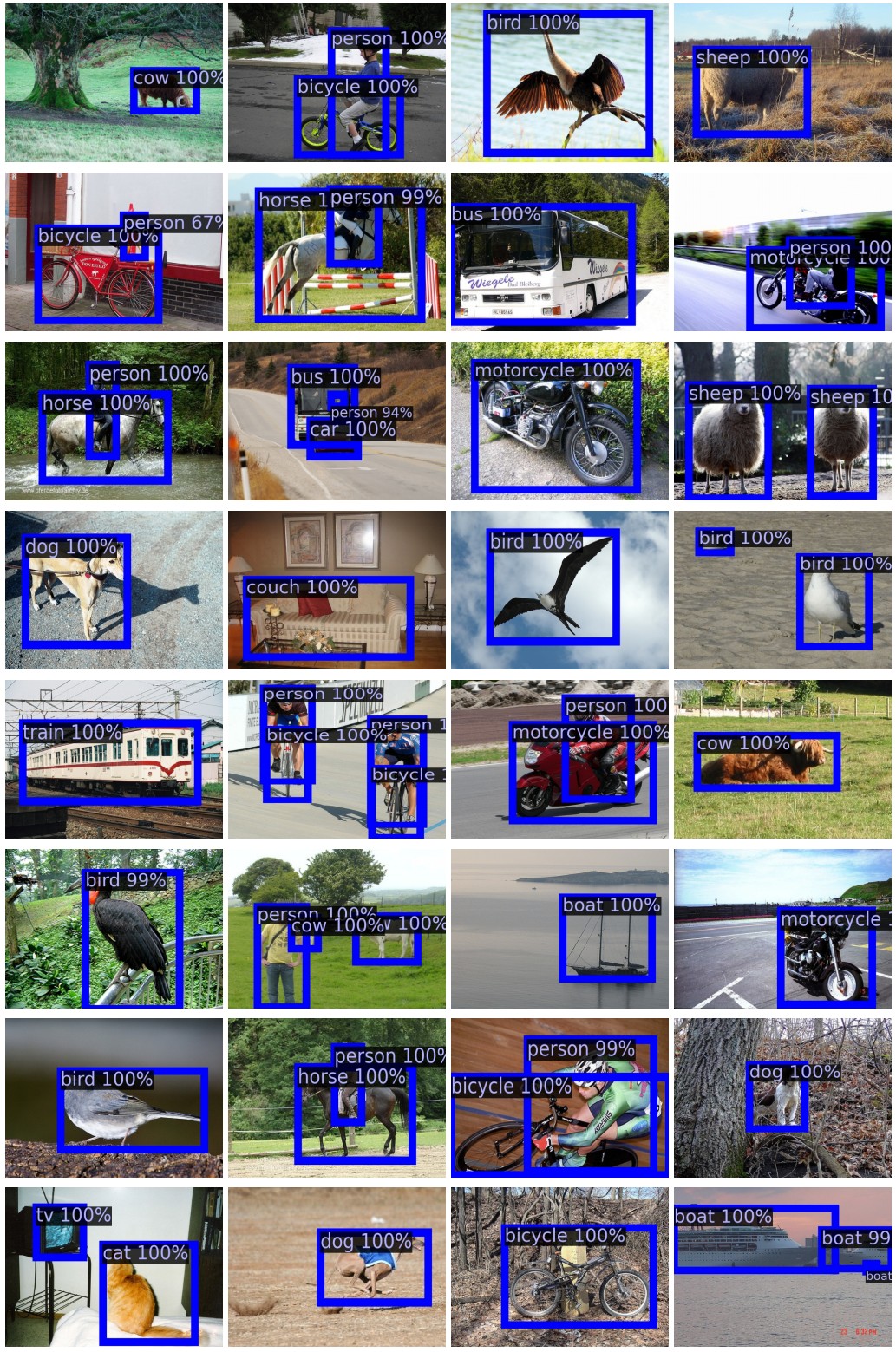

Figure 7: Detection results based on PASCAL VOC. We can see that our method accurately localizes and recognizes objects in these images, e.g., the dog, train, cow, and person, which shows that our method is effective for in-distribution data.

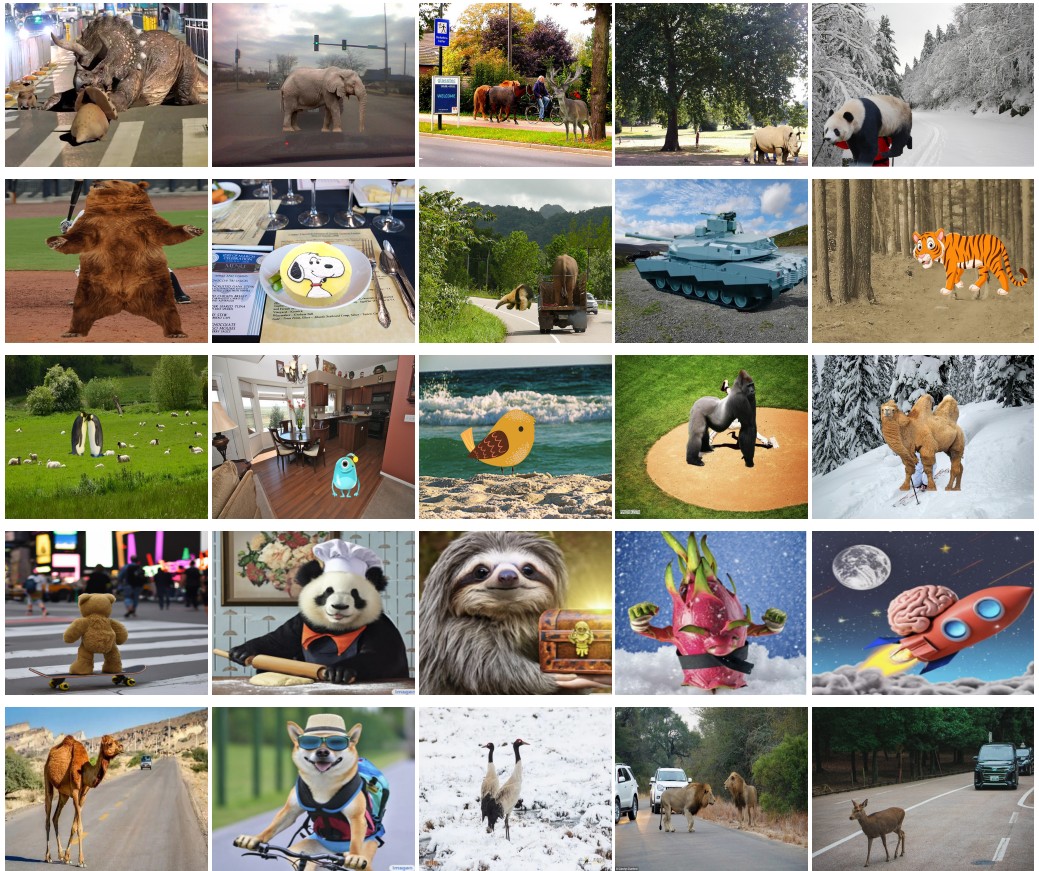

Figure 8: To further evaluate our method, we synthesize some images that contain OOD classes, e.g., the Dinosaur, Panda, and Camel. Meanwhile, we also collect some images from real scenarios.

Table 6: Definitions of notations used in our method.

| Notations | Definition |
|---|---|
| $F$ | The feature map extracted by the backbone network. |
| $Z$ | The encoded Instance map. |
| $P_{\text{in}}$ | The ID object features extracted based on $Z$. |
| $Q_i$ | The output encoding result of the $i$-th IB branch for information enhancement. |
| $G_i$ | The calculated gating weight of the $i$-th IB branch. |
| $A$ | The aggregated results. |
| $E$ | The output enhanced results. |
| $R$ | The encoded OOD map. |
| $P_{\text{ood}}$ | The OOD object features extracted based on $R$. |

## A.8 MORE ABLATION STUDIES ABOUT EQUATION 4

In equation 4, $A$ is to aggregate the results of each IB branch. By this operation, $A$ contains plentiful object-related information. The learned sigmoid weight $\alpha$ is to balance $A$ and $P_{\text{in}}$ during the enhancing process.

We make an ablation analysis about $A$ and $\alpha$. Firstly, we replace the gating operation (as shown in the left part of equation 4) with the simple mean operation and keep other modules unchanged. We observe that compared with our method, the mean operation increases FPR95 by around 2.7%, which shows the effectiveness of the gating operation. Next, we replace the learned sigmoid weight $\alpha$ with a manually set value and keep other modules unchanged. We set multiple different values and observe that 0.6 corresponds to the best performance. However, compared with our method, this

operation increases FPR95 by around 1.3%, which indicates that using the learned weight is much better for balancing $A$ and $P_{\text{in}}$.

## A.9 DEFINITIONS OF NOTATIONS

Table 6 gives the definitions of notations used in our method.

## A.10 LIMITATION

To promote the safe deployment of object detectors, we propose to use information bottlenecks to strengthen object-related information and generate simulative OOD features to alleviate the impact of lacking OOD data for training. Since we only verify our method on two benchmarks, we do not clear about the performance of the proposed method in practical scenes, which may be the limitation of our proposed method.

