# OpenReview forum: "TIB: Detecting Unknown Objects via Two-Stream Information Bottleneck"
_ICLR.cc/2023/Conference — Submitted to ICLR 2023_

### Official Review · Reviewer_ajPF · 2022-10-24

**Confidence:** 4
**Correctness:** 3
**Technical Novelty And Significance:** 3
**Empirical Novelty And Significance:** 2
**Recommendation:** 5

**Clarity, Quality, Novelty And Reproducibility:**

+ The paper's structure is generally well-organized. The problem and the motivation are well described. However, there are some problems with the clarity and quality, i.e., the authors use many notations without clearly summarizing or referring back to Figure 2. It would be much better to follow if the authors include a summarized table of notations and add the notation, e.g. F, P_in​, E, ….  to Figure 2.
+ A limitation section should be included.
+ The method of applying IB to out-of-distribution appears to be novel. The designs of TIB and RIB could also be useful to the community since they could be easily applied to other object detection problems.


**Strength And Weaknesses:**

**Strengths**
+ The idea of using IB to extract object-related and OOD feature problems is novel and interesting. It seems that the idea could be applied to other object detection problems. The empirical results are encouraging.
+ The paper is well-written and easy to follow. The key design choices in the paper are carefully described.

**Weaknesses**
+ The task of OOD-OD is not formally described in the paper.
+ Does background differ from OOD instances?
+ Is the OOD bounding box included when performing inference?
+ The correlation between Information Bottleneck and the method is not clearly illustrated. Particularly, the KL terms in Eq. (5) do not express the IB where Z should have low mutual information with F but ignore the high mutual information with the label. So do the Q and P_in. Eq. (8) also has that problem. Also, we usually minimize a loss (with a non-negative value) not maximize a loss as in Eq. (8).
+ The enhancing process (Eq. 4) appears to be unclear. Both A and α are related to P_in, and the role of each term is not well described. An ablation study about this process may help to clarify.
+ Regarding the datasets, how many classes are in the ID and OOD data?
+ Images in Figure 4 only contain objects from ID classes, which makes the visualization of OOD maps uninformative. What do the OOD maps look like with objects from  OOD classes, and in the cases of mixed ID and OOD classes? It would definitely demonstrate the effectiveness of the proposed approach on OOD-OD.
+ There is no discussion about the choice of hyper-parameters in Section A.3. The paper should include an explanation for each set of hyper-parameters.


**Summary Of The Paper:**

This paper addresses the problem of out-of-distribution object detection (OOD-OD), where an object detector has to detect unknown objects without relying on an auxiliary dataset. The authors propose a framework of Two-Stream Information Bottleneck (TIB), with a standard IB network to extract object-related information from in-distribution (ID) data, and a Reverse Information Bottleneck (RIB) to obtain OOD features. By exploiting IB, the method manages to obtain enhanced object-related information from ID data. At the same time, TIB obtains simulative OOD features by maximizing the discrepancy with the ID features. Experimental results show the method could significantly improve the performance of OOD-OD and incremental object detection over the baselines.

**Summary Of The Review:**

The idea in the paper is well-motivated and described. The paper addresses the problem of out-of-distribution object detection by exploiting Information Bottleneck, which has been carefully studied by prior work and shown promising results. Therefore, the idea has solid theoretical reasoning. However, when implementing the IB in terms of loss functions, not all properties are guaranteed. Also, lacking of a visual explanation of the OOD map with OOD instances significantly reduces the convincingness of the proposed approach.

---

> ### Author Response · Authors · 2022-11-09
> **Thanks for your helpful comments. We will modify our paper carefully.**
>
> 1. Formal Description of OOD-OD:
>
> We have modified our paper and given the formal description of OOD-OD in Sec. 3.
>
> For unsupervised OOD-OD, the object detector is trained based on the ID data {$X$, $Y$, $B$}, where $X$ denotes the set of ID images, $Y$ is the label set, and $B$ indicates location information. During inference, give an image x* including OOD objects, the trained object detector should accurately distinguish ID objects (the output is 1) from OOD objects (the output is 0).
>
>
> 2. The Relation between Background and OOD Objects:
>
> For an object detector trained based on close-set assumption, the background is often defined as the content unrelated to the known categories. Taking the input image in Fig. 2 as an example, the background contains rich information, e.g., the road, and tree. OOD objects could be defined as foreground instances in the background, e.g., the deer. Thus, OOD-OD aims to detect unknown objects in the background.
>
> 3. On OOD Bounding Box during Inference:
>
> During inference, we do not need to calculate OOD bounding boxes based on the simulative OOD map.
>
> 4. More Interpretations about Eq. (5) and (8):
>
> (1) In general, mutual information aims to represent the correlation between two variables. For the representation $Z$, mutual information is to reduce the correlation between $Z$ and $F$, and promote $Z$ to contain rich object-related information. In Eq. (5), since the prior distribution $r$ is unrelated to $F$, minimizing KL loss equals narrowing the relevance between $Z$ and $F$. Meanwhile, minimizing the loss L_cls calculated based on $Z$ is helpful for promoting $Z$ to contain plentiful object-related information. The second KL term in Eq. (5) is also to reduce the relevance between $Q$ and $P$.
>
> (2) As analyzed in Eq. (7), maximizing the loss L_dis in Eq. (8) is to enlarge the gap between the simulative OOD features and ID features, which is helpful for promoting the simulative features to contain sufficient OOD information.
>
> 5. More Ablation Studies about Eq. (4):
>
> In Eq. (4), $A$ is to aggregate the results of each IB branch. By this operation, $A$ contains plentiful object-related information. The learned sigmoid weight $\alpha$ is to balance $A$ and $P$ during the enhancing process.
>
> We make an ablation analysis about $A$ and $\alpha$. Firstly, we replace the gating operation (as shown in the left part of Eq. (4)) with the simple mean operation and keep other modules unchanged. We observe that compared with our method, the mean operation increases FPR95 by around 2.7%, which shows the effectiveness of the gating operation. Next, we replace the learned sigmoid weight $\alpha$ with a manually set value and keep other modules unchanged. We set multiple different values and observe that 0.6 corresponds to the best performance. However, compared with our method, this operation increases FPR95 by around 1.3%, which indicates that using the learned weight is much better for balancing $A$ and $P$.
>
> We have added these analyses in Sec. A.8 of the Appendix.
>
> 6. On the Classes of ID and OOD Data:
>
> For ID data, PASCAL VOC and BDD-100k separately contain 20 and 10 classes. In general, COCO and OpenImages respectively include 80 and 600 classes. And OOD data contains disjoint labels from the respective ID dataset.
>
> 7. On Fig. 4:
>
> The images in Fig. 4 do not contain ID categories. The ID categories are Person, Car, Bicycle, Boat, Bus, Motorbike, Train, Airplane, Chair, Bottle, Dining Table, Potted Plant, TV, Sofa, Bird, Cat, Cow, Dog, Horse, and Sheep.
>
> Due to lacking unknown data during training, the simulative OOD map only aims to deviate from the current instance features significantly. Thus, for OOD classes and mixed ID and OOD classes, the generated simulative OOD maps all deviate from the current instance features. Visualization results in Fig. 4 show that our method is indeed helpful for obtaining effective OOD features, which improves the performance of distinguishing OOD objects.
>
> 8. More Explanations about Hyper-Parameters:
>
> Our method mainly contains three hyper-parameters $\beta$, $\lambda$, and $\tau$. Particularly, since the loss L_cls, L_loc, and L_uncertainty are directly related to the task, $\beta$ and $\lambda$ should be set to a smaller value than $\tau$. Meanwhile, if $\beta$ and $\lambda$ are set to a very small value, KL-loss and L_dis may be weakened in optimization. Thus, it is meaningful to set proper values for these hyper-parameters.
>
> We have added these analyses in Sec. A.3 of the Appendix.
>
> 9. On Notations:
>
> We have given a summarized table of notations in Sec. A.9 of the Appendix. Meanwhile, we have added the notations to Fig. 2.
>
> 10. On Limitation:
>
> We have added a limitation section in Sec. A.10 of the Appendix.

---

> > ### Comment · Reviewer_ajPF · 2022-11-18
> > **Response to the authors**
> >
> > First, I would like to thank the authors for your response. I still have some concerns about the loose connection between the Information Bottleneck theory, i.e., the mutual information in Eq. (1) and the realization of it in Eq. (5) and Eq. (8). It seems like the authors tried to fix a theory to their workable loss rather than derive the theory to the loss of the paper. Furthermore, in the OOD features of Fig. 4, I consent with other reviewers that the OOD features are uninformative, which cannot explain why the RIB works.

---

> > > ### Author Response · Authors · 2022-11-18
> > > **Thanks for your helpful comments.**
> > >
> > > 1. On the Mutual Information:
> > >
> > > In this paper, we follow the idea of deep variational information bottleneck [1] and explore utilizing KL-divergence to approximate the mutual information. Particularly, in Eq. (5), by minimizing the KL-divergence, the intermediate representations can be promoted to contain plentiful object-related information, which is beneficial for improving the detection performance. Meanwhile, the loss in Eq. (8) equals to enlarge the mutual information between the simulative OOD features and the features for classification, which is instrumental in constraining the simulative OOD features to contain rich object-irrelevant information. Thus, our method follows the theories of information bottlenecks and does not fix theories to our loss.
> > >
> > > [1] Alemi, Alexander A., et al. "Deep variational information bottleneck." ICLR, 2017.
> > >
> > >
> > > 2. Further Discussion about Fig. 4:
> > >
> > > The OOD features are informative and could explain the effectiveness of the RIB method. Due to lacking OOD data for training, we can only enlarge the gap between the OOD features and the real classification features, which is helpful for ensuring the generated features contain OOD-related information. Obviously, in Fig. 4, the generated OOD features are different from the instance features, which shows that our RIB method is indeed helpful for obtaining the features containing plentiful information deviating from instance features. Besides, A.1 in Appendix further indicates the proposed RIB could generate features involving rich OOD-related information.

---

### Official Review · Reviewer_UuoW · 2022-10-24

**Confidence:** 5
**Correctness:** 4
**Technical Novelty And Significance:** 3
**Empirical Novelty And Significance:** 3
**Recommendation:** 6

**Clarity, Quality, Novelty And Reproducibility:**

-The writing clarity is really good and the idea of leveraging information bottleneck to enhance object relevant information and suppress object irrelevant information is novel.
- With a few additional details asked above being answered, I believe a skilled researcher/engineer should be able to reproduce this work.
-  I believe with a few minor revisions, the paper is of great value to the conference.

**Strength And Weaknesses:**

Strengths
- The proposed method is very simple and effective and is one of the major strengths of this work.
- The use of information theoretic methods, the IB, for object detection is new and fresh.
- All the claims made in the paper have been properly addressed with adequate experiments.
- The paper tackles a very important practical problem with object detectors and achieves impressive performance on standard benchmarks.

Weaknesses
- I believe the comparisons in Table-1 are unfair as the current method introduces more trainable parameters than VOS. Not to diminish the impressive performance, I believe it is imperative to add a column with backbones and trainable parameters introduced for a more fair comparison.
- Authors claim that the OOD map simulates outlier samples and the visualizations of the map doesn't seem to do that. After careful observation one can notice that the maximum of the OOD map still occurs either inside or around the object of interest. Additionally, these maps are not interpretable, it is not clear to me why these maps should look like noise. Can these maps be interpreted as adversarial noise (on the features) since they are obtained by maximizing the discrepancy between logits of the right class? (Like noise that can be added to make the feature lie closer to a different ID class different than the one in the feature map).
- In the standard OOD-OD setup the number of OOD samples coming from COCO/Open-Images are usually less than 20% (930 OD samples for COCO with 4952 samples of ID val set). It is not clear how the proposed model does with higher levels of contamination from OOD classes.
- The contribution of the first stage of IB is not clear from the ablation experiments. What happens if one were to simply pass F through two different convolutional blocks and keep everything the same? In Table 4, authors either keep the full branch or remove it entirely, but why do one need the first IB/RIB stage? One can apply the same L_dis, L_uncertainty losses on this architecture even without computing mean and standard deviation and combining them. I understand that the loss becomes uninterpretable but how does it fare against the proposed approach?

Minor questions/comments to improve the paper (Not dependent on the final decision):
- It is not clear how Eq. 5 is related to Variational Information Bottleneck. From the paper there should be a negated term and the second term is the KL divergence. In Eq. 5 there are two KL divergence terms and it is not clear how the authors end up with this equation. This step requires some more information to help the readers understand the math.
- How is Eq. 8 related to Eq. 7. Do the authors mean to say that Eq. 8 approximates Eq. 7. In that case why is it hard to compute Eq. 7 directly without any approximation?
- It would be beneficial to researchers implementing this work, if information about computing mutual information (I(.,.)) on 3d feature maps is provided in the supplementary.
- Is the purpose of experiments on Incremental object detection just to show that IB can improve overall object detection performance?. If uncertainty loss is not computed for these experiments, then why would one need to perform RIB?
- Kindly add the baseline to Table 4, readers would have to move back and forth to understand what the baseline is.
- Line before "Metrics" in Section 4.1 ".... manually examined the OOD images to ensure .." -> ".... manually examined to ensure ... ".
- What is the confidence threshold used to generate the results in Fig. 3. If its the same value for VOS and TIB?


**Summary Of The Paper:**

- This paper proposed a new information theoretic approach to tackle out-of-distribution object detection (OOD-OD).
- Authors leverage Variational information bottleneck (IB) to suppress object-irrelevant representations to improve discriminability of in-distribution class (ID).
- Next, to simulate OOD features to train an OOD detector in an unsupervised fashion, authors reverse the IB equation, called Reverse Information Bottleneck (RIB). In contrast to IB, which aims to extract task-relevant compact codewords/features, the goal of RIB is to simulate task-irrelevant OOD style features to train a strong OOD detector.
- Extensive experiments on standard OOD-OD datasets and incremental object detection datasets show the effectiveness of the proposed approach for OOD-OD.

**Summary Of The Review:**

- Overall I'm very happy with the paper and its writing quality.
- Authors leverage Information bottleneck to simultaneously enhance object relatedness and suppress (with reversal) object irrelevance in features to improve OOD-OD.
- The proposed method is simple and very effective. The experiments support all the claims made by the authors. A few more explanations to understand the working of Reverse IB are required which I believe the authors can provide in reasonable time.
- I'm willing to improve my ratings depending on the author's response.

---

> ### Author Response · Authors · 2022-11-08
> **More OOD images are selected and synthesized to verify our method.**
>
> 1. The Comparisons in Table 1:
>
> We have modified Table 1 and added a column with backbones and trainable parameters.
>
> 2. About the Interpretation of the OOD Map:
>
> (1) We agree with you that the maximum of the OOD map occurs either inside or around the object of interest. The reason is that the OOD map and Instance map exploit the same object proposals calculated based on the Instance map to maximize the prediction discrepancy (Eq. (8)). This operation is helpful for enlarging the object-level gap between the OOD map and the Instance map. Thus, it is reasonable that the maximum of the OOD map occurs either inside or around the object of interest, which further shows that our method is beneficial for improving the ability of detecting OOD objects.
>
> (2) It is inappropriate to interpret the OOD map as adversarial noise. In general, adversarial noise aims to add perturbation to misclassify the current objects into a certain different ID category. This operation may weaken the discrimination ability of ID objects. Meanwhile, this operation does not produce the OOD features, which could not improve the performance of distinguishing OOD objects. Here, we make an experiment that takes the feature maps adding adversarial noise as the OOD map. Based on FPR95, compared with VOS, the performance is increased by around 5.1%.
>
> The OOD map could be interpreted as containing plentiful information that deviates from the Instance feature distribution. Concretely, we separately employ the convolution operation (Eq. (2) and (6)) to estimate the distribution information of each element in the extracted feature map. By maximizing the prediction discrepancy, it is helpful for enlarging the distribution gap between the OOD map and Instance map from the background and foreground perspectives.
>
> 3. The Performance on More OOD Samples:
>
> We select 8K images from COCO and synthesize some OOD objects on these images to perform a further evaluation. In Fig. 8 of the Appendix, we show some synthesized images. Through experiments, we observe that our method still outperforms VOS significantly. Particularly, compared with VOS, our method reduces FPR95 by around 10.1% and improves AUROC by around 3.2%, which further demonstrates the effectiveness of our method.
>
> 4. Ablation Analysis of IB:
>
> (1) We observe that if simply passing $F$ through two different convolutional blocks and keeping other modules unchanged, the performance decreases severely. Specifically, based on FPR95, compared with our IB method, the value increases by around 5.3%.  Based on AUROC, the value decreases by around 1.8%. This shows that using IB is indeed helpful for purifying object-related information and improving detection performance.
>
> (2) We agree with you that if only using L_dis and L_uncertainty, the interpretation of these two losses will be weakened. In this paper, we make extensive experiments and observe that estimating means and variances is important for our method. Particularly, compared with our method, the FPR95 value of only using L_dis and L_uncertainty increases by around 9.2%. The AUROC value decreases by around 2.7%.
>
> 5. Interpretation about Eq. (5):
>
> The loss in Eq. (5) corresponds to the top branch of Fig. 2, which aims to extract object-related information. Particularly, there are two KL divergence terms in Eq. (5). The first KL divergence term aims to constrain the encoded Instance map $Z$ to contain rich object-related information. Similarly, the second KL divergence term is to further purify object-related information in the extracted object proposals. By means of these two KL terms, the discrimination ability of the object detector could be improved effectively.
>
> 6. The Relation between Eq. (7) and (8):
>
> Eq. (8) is an approximation of Eq. (7). And Eq. (7) can be directly calculated. However, we observe that using Eq. (8) is much better for enlarging the gap between the object-level features from the OOD map and that from the Instance map, which promotes the OOD map to contain plentiful object-irrelevant information. Here, we make an ablation analysis. Compared with calculating Eq. (8), computing Eq. (7) increases FPR95 by around 1.5%.
>
> 7. Computation of Mutual Information:
>
> We mainly employ KL-divergence to approximate mutual information (Eq. (5)). Sec. A.6 of Appendix has given the details.
>
> 8. The Purpose of Incremental Object Detection:
>
> The purpose of Incremental Object Detection is to show that our method is beneficial for improving the discrimination ability of the object detector. Concretely, by using the IB operation, the object-related information in the extracted features could be enhanced. Besides, we also use the RIB branch to generate simulative features. By recognizing the simulative features as the background category, the discrimination ability of the object classifier could be effectively strengthened.
>
> 9. The Confidence Threshold:
>
> In Fig. 3, the confidence threshold is set to 0.95, which is the same as VOS.

---

> > ### Comment · Reviewer_UuoW · 2022-11-09
> > **Clarification required on a few comments**
> >
> > I thank the authors for their detailed response.I don't see the changes made by the author to the paper (In particular the Fig. 8 that the authors were talking about). Kindly update the paper at your convenience.
> >
> > 3. Can the authors describe the details of the experiments for this experiment? Do the authors mean that the validation set consists of 4952 samples from the ID set and 8000 samples from the OOD set? What is the ratio of number of ID instances (bounding boxes) to the OOD instances in this validation set? How does the model perform at higher levels of contamination?
> >
> > 8. If there is no uncertainty loss, and there is no OOD detector, what is the RIB branch trained with?

---

> > > ### Author Response · Authors · 2022-11-10
> > > **Thanks for your helpful comments**
> > >
> > > We will update the paper as soon as possible.
> > >
> > >
> > > On the Details of the New Experiment:
> > >
> > > In the new experiment, we use PASCAL VOC as the ID data for training. We do not change the training settings. During inference, the validation set consists of 4952 samples from the ID set and 8,000 samples from the OOD set. The ratio of the number of ID instances to the OOD instances in this validation set is around 1.0 : 2.3. To improve the complexity of the testing scene, we synthesize some OOD objects into real images. For example, we add some real and cartoon animals, e.g., elephants, lions, and dinosaurs.
> > >
> > > On the RIB branch:
> > >
> > > If there is no uncertainty loss, and there is no OOD detector, we can use two different ways for this case:
> > >
> > > The first method is that we directly take the synthesized features as the background features to calculate the corresponding loss, which equals performing an augmentation of the background.
> > >
> > > The second method is to train a binarized classifier, i.e., the output of the known category is 1, and the output of the synthesized features is 0. By minimizing the cross-entropy loss, the discrimination ability of the object classifier could be enhanced effectively.
> > >
> > > Through experiments, we observe that these two methods could all improve performance. And the performance of the second method outperforms the first method.

---

> > > > ### Author Response · Authors · 2022-11-12
> > > > **We have updated the paper.**
> > > >
> > > > Thanks for your helpful comments. We have updated this paper. We will modify our paper carefully according to your valuable comments.

---

> > ### Comment · Reviewer_UuoW · 2022-11-22
> > **Concerns about the information bottleneck theory and the loss**
> >
> > Hi,
> > I agree with the other reviewer that there is a very loose connection between the loss used in this work and the information bottleneck theory which uses mutual information. I think what would help is to show the original formulation and the realization of the formulation behave similarly during training. Something along the lines of plotting both the values during the whole training process to see if the realization is bounded by the actual formulation.

---

> > > ### Author Response · Authors · 2022-11-22
> > > **Comparison between the information bottleneck theory and our loss**
> > >
> > > Thanks for your valuable advice. To demonstrate the connection between the loss used in our work and the standard mutual information loss, we follow Mutual Information Neural Estimation (MINE) [1] to estimate mutual information. And other modules are kept unchanged. We take PASCAL VOC as ID data for training.
> > >
> > > We show some total loss values every 2,000 iterations.
> > >
> > > Mutual Information Loss: 0.7983 (iter0), 0.4199 (iter1999), 0.3815 (iter3999), 0.3787 (iter5999), 0.3494 (iter7999), 0.3224 (iter9999), 0.2906 (iter11999), 0.2715 (iter13999), 0.277 (iter15999), 0.2973 (iter17999)
> > >
> > > Our Loss: 0.7818 (iter0), 0.4223 (iter1999), 0.3703 (iter3999), 0.3811 (iter5999), 0.3602 (iter7999), 0.3162 (iter9999), 0.2913 (iter11999), 0.25 (iter13999), 0.2674 (iter15999), 0.2763 (iter17999)
> > >
> > > We find that the loss difference is not significant. Besides, the performance of using the standard mutual information loss is slightly lower than our method. We will modify our paper carefully and plot loss curves in our paper.
> > >
> > > [1] Ishmael Belghazi, Aristide Baratin, Sai Rajeswar, Sherjil Ozair, Yoshua Bengio, Aaron Courville, and R Devon Hjelm. "Mine: mutual information neural estimation." ICML, 2018.

---

### Official Review · Reviewer_D2nY · 2022-10-27

**Confidence:** 4
**Correctness:** 3
**Technical Novelty And Significance:** 3
**Empirical Novelty And Significance:** 2
**Recommendation:** 6

**Clarity, Quality, Novelty And Reproducibility:**

The paper is well written, but packed with equations. It would be better to introduce more figures to increase the clarity.

**Strength And Weaknesses:**

Strength
+ The motivation is clear, and the solution is novel.
+ Good performance gain on incremental object detection setup.

Weakness
- The major concern is the scalability of such method. The author only uses PascalVOC and BDD as in-domain data, which is relatively small and simple. To verify its ability of handling real-world data, I would suggest the author can demonstrate a more complex setup, for example, use COCO as ID and Openimages as evaluation set.
- Figure 2 is not very illustrative. It would be better to align equations into the flow. Or teardown to multiple small figures for better clarity.
- In Figure 4, the visualization of OOD is confusing. It is more like white noise. How does simulate OOD feature extracted from it help?

**Summary Of The Paper:**

This paper attempts to solve OOD problem in object detection.
It tries to reduce the impact of lacking unknown data for supervision and leverage in-distribution data to improve the model’s discrimination ability with both Information Bottleneck and Reverse Information Bottleneck.
It first uses a standard IB network to disentangle instance representations that are beneficial for localizing and recognizing objects, and then use a RIB to obtain simulative OOD features to alleviate the impact of lacking unknown data.
Experiments demonstrate good performance in incremental object detection.

**Summary Of The Review:**

Overall, this is a solid paper with detailed experiments and visualizations. The major concern is the scalability in really-world data. I will increase score if the author can provide my evidence.

---

> ### Author Response · Authors · 2022-11-13
> **We have made an experiment to show the scalability of our method.**
>
> 1. On the Scalability of Our Method:
>
> Thanks for your advice. To verify the scalability, we have made an experiment, which uses MS-COCO as the ID data and OpenImages as the OOD data for evaluation. During training, we employ the standard SGD optimizer with a learning rate of 0.02. The iteration number is set to 90,000.
>
> Based on FPR95, AUROC, and mAP, the performance of the baseline method (VOS) is:
>
> 72.91%,      79.04%,        37.3%
>
> The performance of our method is:
>
> 66.43%,      81.93%,        37.6%
>
> We can see that our method still outperforms VOS. Particularly, compared with VOS, our method reduces FPR95 by around 6.48%.
>
> 2. On Fig. 2:
>
> Thanks for your advice. We have modified Fig. 2 and aligned notations into the figure. Meanwhile, in table 6 of the Appendix, we have added a table to give the definitions of notations used in our method.
>
> 3. Further Interpretations about Fig. 4:
>
> Due to lacking unknown data for supervision, to achieve the goal of detecting unknown objects, the generated simulative OOD features are constrained to contain plentiful information that deviates from the current object features. Through this mechanism, the generated features could be considered to involve OOD information.
>
> In Fig. 4, we can see that compared with the Instance map, the simulative OOD map indeed contains sufficient information that deviates from the instance features. This shows that during training, with the help of simulative OOD features, the ability of distinguishing OOD objects could be enhanced effectively.
>
> We agree with you that the visualization results are somewhat like noise. The reason is that our method aims to enlarge the gap between the simulative features and the instance features from background and foreground perspectives, which further shows the advantages of our method.

---

### Author Response · Authors · 2022-11-08
**Thanks to all reviewers**

Thanks for the valuable comments from all reviewers. We will modify our paper carefully according to these comments.

---

### Decision · Program_Chairs · 2023-01-20

**Decision:**

Reject

**Justification For Why Not Higher Score:**

There are some major technical concerns that are not fully addressed in the current manuscript. Please see the meta-review for details.

**Justification For Why Not Lower Score:**

NA

**Metareview: Summary, Strengths And Weaknesses:**

This paper studies a crucial problem of out-of-distribution object detection (OOD-OD). The goal is to detect test-time objects that are outside the known training categories, while at the same time ensuring accurate localization and classification of in-distribution objects. To tackle the problem, this paper introduces a new information-theoretic approach to synthesizing OOD feature maps, which are then used to regularize the training of the object detection network. The novelty behind generating OOD features is based on Reverse Information Bottleneck (RIB), which promotes task-irrelevant information. All reviewers recognize the novelty of the paper, which first connects IB to OOD-OD.

Another strength of the paper is the extensive evaluation and promising performance on the OOD-OD benchmark, for both PASCAL-VOS and BDD datasets as in-distribution. The authors further conducted evaluations on the incremental object recognition task. It might be useful to report results across multiple runs with different random seeds as well.

During the discussion phase, the authors addressed several concerns and improved the clarity of the paper. Some reviewers questioned the soundness of the approach. Specifically, there are two major points raised:

* The gap between mutual information theory and the actual realization in this paper. (the authors have made an attempt to clarify this, with additional experiments)
* The lack of clear insight and evidence of why the proposed method works. In particular, all three reviewers brought up the question of how those obtained OOD feature maps improve the OOD detection performance. To fully understand this, I believe there are two separate analyses needed: (1) The generated OOD map deviates from the instance features, and (2) why training the object detection model with the noise-like OOD feature maps can effectively generalize the detection to future unseen OOD objects, most of which may not necessarily have noise-like patterns. While the authors have provided evidence for (1), but the crucial analysis for (2) is currently missing. Instead of learning the noisy OOD feature maps using RIB, an interesting baseline would be to directly generate noise (with similar magnitude) for performance comparison. It would be important to understand the characteristics of the learned OOD feature maps and how they help with the detection of real OOD objects. A related suggestion made by Reviewer ajPF would be meaningful to include in the revised manuscript as well. In particular, _``what do the OOD maps look like with objects from OOD classes, and in the cases of mixed ID and OOD classes?''_

Overall the reviewers think the paper is in reasonable shape, but the second point raised above does cast technical doubt on the readiness for ICLR. The manuscript will be more convincing after addressing this important concern.